# Social-ecological alignment and ecological conditions in coral reefs

Michele L. Barnes[1,2], Örjan Bodin [ID] [3], Tim R. McClanahan[4], John N. Kittinger [ID] [5,6], Andrew S. Hoey [ID] [1], Orou G. Gaoue[2,7,8,9] & Nicholas A.J. Graham [ID] [1,10]

Complex social-ecological interactions underpin many environmental problems. To help capture this complexity, we advance an interdisciplinary network modeling framework to identify important relationships between people and nature that can influence environmental conditions. Drawing on comprehensive social and ecological data from five coral reef fishing communities in Kenya; including interviews with 648 fishers, underwater visual census data of reef ecosystem condition, and time-series landings data; we show that positive ecological conditions are associated with 'social-ecological network closure' – i.e., fully linked and thus closed network structures between social actors and ecological resources. Our results suggest that when fishers facing common dilemmas form cooperative communication ties with direct resource competitors, they may achieve positive gains in reef fish biomass and functional richness. Our work provides key empirical insight to a growing body of research on social-ecological alignment, and helps to advance an integrative framework that can be applied empirically in different social-ecological contexts.

[1] ARC Centre of Excellence for Coral Reef Studies, James Cook University, Townsville, QLD 4811, Australia. [2] Department of Botany, University of Hawaii at Manoa, 3190 Maile Way, Honolulu, HI 96822, USA. [3] Stockholm Resilience Centre, Stockholm University, 106 91 Stockholm, Sweden. [4] Wildlife Conservation Society, Marine Programs, Bronx, NY 10460, USA. [5] Center for Oceans, Conservation International, Honolulu 96821 HI, USA. [6] Center for Biodiversity Outcomes, Julie Ann Wrigley Global Institute of Sustainability, Arizona State University, Tempe 85281 AZ, USA. [7] Department of Ecology and Evolutionary Biology, University of Tennessee, Knoxville, TN 37996, USA. [8] Faculty of Agronomy, University of Parakou, 01 BP 123 Parakou, Benin. [9] Department of Geography, Environmental Management and Energy Studies, University of Johannesburg, APK Campus, Johannesburg 2092, South Africa. [10] Lancaster Environment Centre, Lancaster University, Lancaster LA1 4YQ, UK. Correspondence and requests for materials should be addressed to M.L.B. (email: Michele.Barnes@jcu.edu.au)

Humans are a fundamental part of ecosystems and rely on them to support a wide array of their needs. The extent of environmental stressors connected to human activities thus makes understanding social–ecological linkages of central importance for the analysis of almost any action related to securing a sustainable future[1]. Recognizing this, research on the environment is increasingly focused on transcending traditional disciplinary boundaries and embracing an integrative, complex systems view to understand ecosystems from a perspective that incorporates theories and frameworks from both the natural and social sciences[2,3]. Even with this progress, studying complex systems involves inherent limitations, including a lack of common language and methods shared between the natural and social sciences[4,5]. Thus advancing tractable and informative frameworks and models that capture social–ecological linkages and can be applied empirically remains a defining challenge to address real-world sustainability issues.

A path forward that is gaining increasing attention in the literature is the development and application of social–ecological network approaches[4,6–9]. Network approaches offer a fruitful framework for theorizing and empirically investigating important social–ecological interactions and how they relate to sustainability outcomes for several reasons. First, social–ecological network approaches can capture important relationships both among and between social and ecological entities (Fig. 1), thus explicitly accounting for interdependencies (e.g., spillovers and feedbacks) that can have dramatic effects on social–ecological system behavior[10]. Second, social–ecological network approaches evoke language, methods, and models common to both the natural and social sciences[11,12], thus providing one avenue to facilitate the cross-disciplinary engagement necessary for solving complex environmental problems. Yet, despite recent theoretical and conceptual developments of social–ecological network approaches[13], empirical applications have struggled to move beyond individual case studies or explicitly link aspects of social–ecological structure to quantitative data on ecosystem conditions[13–15]. We advance this emerging research through a

novel multi-case, comparative empirical assessment that demonstrates how certain social–ecological interdependencies relate to quantitative ecological conditions.

Our research rests on the assumptions that (a) important aspects of social systems, ecological systems, and the interactions between them can be modeled and analyzed as nodes and links in a multilevel social–ecological network, and (b) social–ecological networks are themselves composed of precisely defined network configurations [i.e., building blocks or network motifs[16]] that reflect key relationships among social actors and ecological resources important for achieving particular outcomes (Fig. 1)[4]. Perhaps the most salient social–ecological network configuration highlighted to date[17,18] is the closed, cross-level social–ecological triangle—where two actors connected to the same resource are also connected to each other (Fig. 1). This configuration captures a form of 'social–ecological network closure', i.e., fully linked and thus closed, network structures between social actors and ecological resources (which stand in contrast to open social–ecological network structures; e.g., where social actors are connected to common ecological resources but are not connected to each other). In social network science, network closure[19] [often equated with bonding social capital[20]] emphasizes that tight coupling between actors facilitates trust, learning, and the establishment of common norms and sanctions while minimizing uncertainty[21,22]. Social–ecological network closure extends this coupling across the social–ecological divide, identifying specific forms of communication and cooperation that bind actors connected to the same (or interconnected[23]) resources, thereby better equipping them to learn from each other and agree on and address important environmental problems (Fig. 2).

The proposed utility of this type of social–ecological network closure is especially pronounced in the commons, where actors use shared resources for extractive purposes[14]. In this context, actors are faced with a ubiquitous social dilemma, i.e., the tragedy of the commons[24], whereby each individual has an incentive to overharvest in order to maximize their own short-term gain due to the non-excludable and rivalrous nature of common resources.

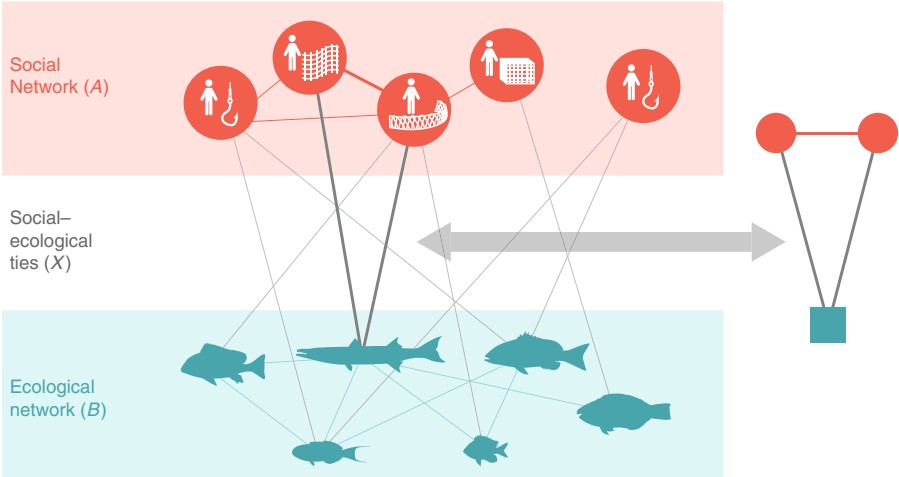

**Fig. 1** A coral reef fishery as a multilevel social–ecological network. An illustrative example of the integrative, social–ecological network modeling approach and key configuration of interest. The social network (A) captures key communication relationships between individual fishers. The ecological network (B) captures trophic interactions among target species. In reef fisheries, each fishing gear type catches a diverse and overlapping, but distinct assemblage of species in B. Individual fishers are thus linked to particular fish species (X; social–ecological ties) depending on the type of gear they use (depicted in the nodes in A). All nodes and links are representative of our empirical data. The multilevel structure (A, B, X) captures the dependencies that exist within the system, i.e., how features of social and ecological systems are interrelated both within and across levels. Full multilevel social–ecological networks can be disassembled into smaller building blocks, or key configurations (right), that form the foundation for the larger system structure[4,68]. Here a form of social–ecological alignment is emphasized, i.e., social–ecological network closure, which captures the tendency for actors tied to the same resource to form cooperative communication ties

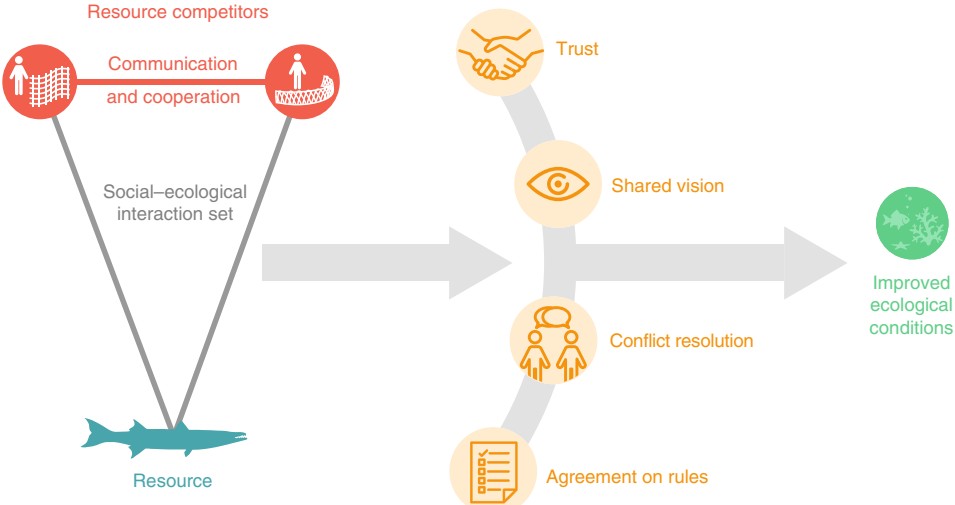

**Fig. 2** Theoretical mechanisms linking social–ecological network closure and ecological conditions. A conceptual diagram illustrating key social processes theoretically supported by social–ecological network closure that may lead to improved ecological conditions in the commons. When direct resource competitors in settings characterized by strong and complex patterns of social–ecological interactions form cooperative communication ties, it lays the foundation for the emergence of trust, a shared vision, and sustained commitments[27,29,30] regarding the management of shared resources. Two examples of such commitments include the development of conflict resolution mechanisms and agreement on rules. These social interactions and processes can ultimately lead to improved ecological conditions. It is important to note that this figure is only illustrative of key mechanisms linking social–ecological network closure to ecological conditions and does not include the full range of social–ecological interactions and feedbacks that can affect both ecological and social conditions in any given environmental system

Privatization or third-party regulation and enforcement can help to solve this dilemma; however, these actions are not always feasible, preferable, or cost effective. In such cases, the ability of resource users to act collectively to devise and enforce commonly agreed upon norms and rules for sustainable resource use is critical[25]. Yet, how such cooperation emerges when faced with social dilemmas without oversight from a central authority has been of considerable interest among scholars for decades[26]. Though several explanations have been proposed and some have been supported through empirical research[27], one of the most robust findings has been that communication is critical—when individuals engage in face-to-face communication, cooperation increases significantly[28]. Thus, if actors with a stake in the same resource have opportunities to communicate, there is strong theoretical evidence to support the notion that it can facilitate cooperation toward effectively managing shared resources, thereby leading to improved ecological conditions (see Fig. 2)[4,27,29,30]. This type of social–ecological network closure can also facilitate learning, which is critical for updating management strategies in the face of social and ecological change[31]. In common-pool resource settings, social–ecological closure is thus an important aspect of what is often referred to as social–ecological alignment (or social–ecological fit) where relationships between social actors are aligned with the characteristics of the underlying biophysical system[14,32].

Coral reef fisheries are an ideal common-pool resource system to investigate the potential utility of this form of social–ecological alignment. Reefs are one of the most productive and biologically diverse ecosystems on the planet[33], providing critical services that support the livelihoods of millions of people[34]. Yet, reefs are rapidly degrading on a global scale[33], in part due to unsustainable fishing[35]. All reef fisheries face (or have faced) the tragedy of the commons, and most are characterized by multiple species being targeted (or incidentally caught) by multiple gears (Fig. 1). This complexity in the resource base (network level *B*, Fig. 1) and associated harvesting strategies (network level *X*, Fig. 1) presents considerable challenges for sustainable management[36]. Most coral

reefs are also located in regions that suffer from low institutional capacity for governance, high dependence on reef resources, and high rates of poverty[37]. Thus a better understanding of how social–ecological alignment relates to ecological conditions in coral reef fisheries could potentially have large implications for millions of people worldwide.

Here we test the hypothesis that social–ecological network closure is associated with positive ecological conditions in the face of the commons dilemma. We do so by examining whether cooperative communication relationships between fishers harvesting the same species (i.e., closed, cross-level social–ecological triangles, Fig. 1) mediate biomass and functional richness of fished resources across five coral reef fishing communities (sites) along the Kenyan coast (Methods). Our ecological indicators— reef fish biomass and functional richness—are strong predictors of reef ecosystem condition. Reef fish are key elements of reef ecosystems that drive processes linked to ecosystem condition and stability[38]. Fish biomass has been shown to be related to a wide range of information on reef fish functioning (e.g., herbivory, predation), trophic structure, life history composition, and benthic ecosystem state[39,40]. The magnitude of fishable biomass is highly sensitive to fishing and is commonly used to gauge the status of coral reefs globally[41]. Functional richness captures the roles species perform in an ecosystem by categorizing species based on a combination of key traits (e.g., diet, body size, and mobility), rather than taxonomy. As such, functional richness quantifies the number of unique trait combinations within a given sample and has been shown to predict ecological responses to disturbance, understand competitive interactions, and partly drive productivity[42]. Functional, as opposed to taxonomic, richness is fast becoming a much preferred measure of biodiversity in ecology as it captures more about the role of species in ecosystem functioning[42,43].

To support our inquiry of the role of social–ecological network closure on ecological conditions in reef fishing communities, we accounted for biophysical, environmental, and human impact characteristics known to effect reef ecosystem conditions. We also

| | Concept | Site | | | | |
|---|---|---|---|---|---|---|
| | | A | B | C | D | E |
| Social–ecological network closure | | 0.07 (0.02)* | 0.08 (0.03)* | 0.08 (0.02)* | 0.06 (0.03) | 0.04 (0.03) |
| Social network density | | −7.84 (0.36)* | −6.38 (0.88)* | −7.60 (0.29)* | −7.31 (0.59)* | −6.06 (0.71)* |
| Social network centralization | | 0.00 (0.11) | 0.20 (0.21) | 0.29 (0.09)* | 0.13 (0.19) | −0.30 (0.22) |
| Social network closure | | 0.68 (0.10)* | 0.44 (0.13)* | 0.63 (0.10)* | 0.61 (0.19)* | 0.45 (0.17)* |
| Leader activity | | 0.82 (0.18)* | 0.98 (0.19)* | 0.84 (0.16)* | 1.67 (0.40)* | 1.45 (0.31)* |
| Landing site homophily | | 3.14 (0.23)* | 1.18 (0.62) | 1.73 (0.15)* | 2.61 (0.40)* | 2.50 (0.35)* |

**Fig. 3** The importance of social–ecological network closure. Values shown are the coefficients (and SEs) of social–ecological network closure (shaded) and other key parameters from five multilevel exponential random graph models (ERGMs) fit to empirical social–ecological networks representing each of the five reef fishing communities studied (sites A–E). Shapes and colors in the conceptual graphical depictions follow Fig. 1. *L* indicates an actor in the social network who is also a leader, and the tie linking this leader to another social actor demonstrates the potential for leaders to have more ties on average than others; *a* indicates an actor in the social network who uses hypothetical landing site *a*, and the tie linking this actor to another whom also uses landing site *a* demonstrates the potential homophily effect on landing site. Note that the depictions for centralization and closure in the social network are only representative and do not explicitly capture the alternating nature of the specific parameters included in the model (termed ASA and ATA in MPNet; Methods, Supplementary Methods). Full models also included controls for activity in each landing site where a residual analysis suggested that fishers may be more active in forming and maintaining ties than would be expected by chance alone (Supplementary Table 5). Asterisk (*) indicates significance at *P* < 0.05

evaluated other social and institutional conditions known to effect collective management of the commons to determine whether they provided alternative explanations for the relative ecological condition of some sites versus others (Methods). Finally, we conducted a preliminary assessment of indicators of the key social processes supported by social–ecological network closure (Fig. 2) across sites to explore whether they aligned with our theoretical expectations. Taken together, our results provide support to our hypothesis that social–ecological network closure is associated with positive ecological conditions. Specifically, our results indicate that when fishers facing commons dilemmas form cooperative communication ties with direct resource competitors, they may achieve positive gains in reef fish biomass and functional richness.

## Results

**Social–ecological ties**. We constructed full, multilevel social–ecological networks akin to Fig. 1 for each reef fishing community, or site (Methods). Across sites, there were 71–232 fishers in each social network (Supplementary Table 1). On average, fishers had 1.52–3.49 contacts with whom they had formed cooperative communication ties specific to fishing and fishery management (i.e., social ties in *A*, Fig. 1). Social–ecological ties (*X*, Fig. 1) linked fishers to their respective target species via the primary fishing gear they used (Methods, Supplementary Methods, Supplementary Tables 2–4). We found at least three, but up to five different types of primary fishing gear in use, which included hook and line, gillnets, seine nets, spears, and traps (Supplementary Table 2). There was substantial—but not complete—overlap in target species across gear types, with the majority of catch from all gear types comprising a total of 36 species (Supplementary Table 3). Many individual fishers thus competed for the same resources, irrespective of their choice of fishing gear (Supplementary Table 4).

**Social–ecological network closure**. We tested whether and to what extent social-ecological network closure helped to explain

the structure of our empirically observed social–ecological networks by leveraging advances in multilevel exponential random graph models[44] (ERGMs; see Methods, Supplementary Methods). We found a significant positive effect of social–ecological network closure in three of our five sites: sites A–C, as indicated by the positive and significant parameter estimates for the closed, cross-level social–ecological triangle (Fig. 3). Thus, in sites A–C, fishers harvesting the same resources were significantly more likely to have formed cooperative communication ties, whereas in sites D and E, they were not. Aside from this effect, results from our ERGMs showed little to no difference across sites in endogenous and exogenous factors structuring the empirical social–ecological networks. In all sites, fishers had a similar baseline tendency to form social ties (social network density, Fig. 3). There was no consistent, significant effect of preferential attachment[45] (centralization) in the social networks (Fig. 3). Fishers had a tendency to form ties with community leaders more so than others in all sites[46], as indicated by the positive and significant parameter estimates for leader activity shown in Fig. 3. There was also a significant homophily effect[47] on landing site in all of our study sites where more than one landing site was in regular use (Supplementary Methods), meaning that fishers tended to preferentially form ties with others from their community who visit the same location to land and sell their fish (Fig. 3). Lastly, we found a significant, positive effect of social network closure[19] (i.e., closure in the social network *A*, Fig. 1), indicating that in all of our sites, there was a general tendency for fishers to form triadic social structures (i.e., a friend of my friend is also my friend; Fig. 3). Importantly, even when controlling for this general tendency for cooperative, triadic structures to emerge in the social network, fishers in only three of our five study sites (sites A–C) had specifically formed cooperative communication ties when they shared the same resource more so than expected by chance alone.

**Ecological conditions**. We found evidence that social–ecological network closure is indeed associated with positive ecological

conditions (Fig. 4). Specifically, we found a significantly higher mean level of both reef fish biomass and functional richness in sites with a positive tendency toward social–ecological network closure (sites A–C) compared to those without [biomass: $t(9.49) = 2.09$, $p = .03$; functional richness: $t(12.45) = 3.56$, $p < 0.01$]. Effect size estimates suggest that these differences are meaningful (Cohen's $D$, biomass = 0.89, 90% CI = 0.17, 1.71; Cohen's $D$, functional richness = 1.55, 90% CI = 0.60, 2.50). Importantly, differences in ecological conditions across sites do not appear to be related to other biophysical, environmental, or human impact factors known to be important for driving reef

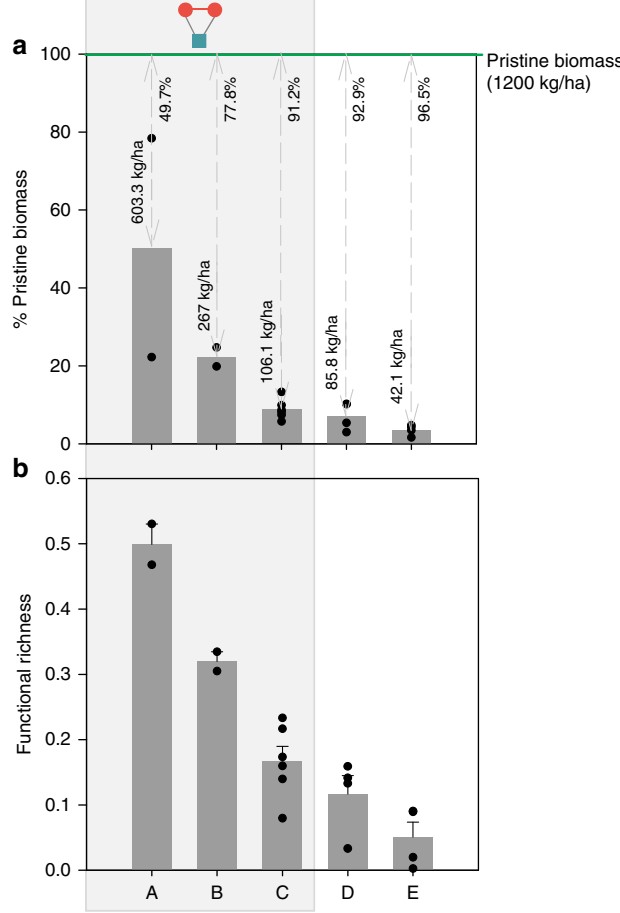

**Fig. 4** Ecological conditions across study sites. Sites that have a significant, positive social–ecological network closure effect (sites A–C) are outlined in the gray box with the network icon. **a** Fish biomass observed in fished areas across each study site from underwater visual surveys compared to the expected level of pristine fish biomass (green line) for unfished reef ecosystems in Kenya, as reported by ref. [69]. Black dots are individual data points; gray bars and text above bars report mean biomass observed; gray arrows denote closeness toward pristine biomass (1200 kg/ha); percentage difference between pristine and observed biomass is reported below the green line. **b** Functional richness of reef fish species (mean ± SE) in fished areas across each study site based on underwater visual surveys and a combination of abundances and trait values. Black dots are individual data points. There is a significantly higher mean level of both reef fish biomass and functional richness in sites with a positive tendency toward social–ecological network closure compared to those without [$t(9.49) = 2.09$, $p = .03$; $t(12.45) = 3.56$, $p < 0.01$; respectively]; and effect size estimates suggest that these differences are meaningful (Cohen's $D = 0.89$, 90% CI = 0.17, 1.71; Cohen's $D = 1.55$, 90% CI = 0.60, 2.50; respectively)

ecosystem conditions (Table 1). Specifically, we found no significant difference between sites with and without social–ecological network closure in terms of sea surface temperature (SST), net primary productivity (NPP), coral cover, rugosity (a measure of structural complexity[48]), human gravity[49] (a human impact measure that accounts for population size and reef accessibility[50]), or fishing pressure (Table 1). The potential differences in shared versus non-shared species comprising our biomass estimates also do not appear to explain these results; e.g., the majority of our biomass estimates are comprised of species that are caught by multiple competing fishers (Supplementary Methods, Supplementary Table 4). These results lend support to our hypothesis that social–ecological network closure can help to overcome commons dilemmas—indeed, where actors linked to the same resource had a significant tendency to form cooperative communication ties (i.e., sites A–C), we saw better ecological conditions.

**Key social processes.** The results of our exploratory assessment of key social processes supported by social–ecological network closure (Fig. 2) partially correspond with our theoretical expectations. First, we found indicative evidence that sites D and E (which do not exhibit a predisposition for social–ecological network closure, Fig. 3) differed from other sites in regard to (1) trust, and (2) shared vision (i.e., resource users have a common understanding of how the system operates and how their actions affect it)[51]. Importantly, we did not find evidence that mean levels of trust differed between sites with and without social–ecological network closure. However, we did find that there was significantly more variation in trust in both sites D and E compared to other sites. This indicates that in sites D and E there is less agreement about whether others can be trusted, and the lack of social–ecological network closure in these sites suggests that there may be pockets of mistrust—or at least a lack of trust—between resource competitors who do not communicate[28]. On the flip side, there may also be pockets of trust. We also found that respondents in site D exhibited significantly more variation in their understanding of the state of coral reef fisheries resources (Table 2). Second, sites D and E also differed from other sites in terms of the commitments made regarding fishery management. For example, in terms of the rules in use, we found that all sites had instituted some form of access rights and designated an area that was closed for fishing. However, only sites A–C had also agreed on and successfully initiated gear restrictions, despite reports that internal conflict over gear use continued to be a problem in both sites D and E. Mechanisms to aid in conflict resolution had also not been designed and established in site E (Table 2).

**Social and institutional conditions.** Success in managing the commons in the absence or failure of top–down governance is known to be associated with a set of social and institutional conditions[25,51,52]. Some of these conditions we argue here are directly supported by social–ecological network closure (e.g., trust, a shared vision; Fig. 2). Yet, others are not (e.g., dependence on common resources; organizational experience/leadership). Thus any variation in these conditions across sites may offer competing explanations for observed differences in ecological conditions. To account for these potentially confounding factors, we used data from our fisher surveys, interviewed community leaders, and drew on existing research[53] (Methods). We found little to no differences across sites in these social and institutional conditions: all had high levels of dependence on fisheries resources, the rights to devise local institutions for management, and had prior organizational experience and local leadership

**Table 1 Biophysical, environmental, and human impact characteristics**

| Metric | Year(s) | With s-e closure (A–C) n; mean (sd) | Without s-e closure (D, E) n; mean (sd) | Two-sample t test t(df) = t value, p value | Effect size Cohen's D [90% CI] |
|---|---|---|---|---|---|
| SST | 2010–2015 | n = 18; 27.33 (0.14) | n = 12; 27.26 (0.13) | t(28) = 1.34, 0.19 | 0.50 [−0.13, 1.12] |
| NPP | 2002–2013 | n = 3; 1021.75 (83.04) | n = 2; 951.77 (0) | t(2) = 1.46, 0.28 | 1.03 [−0.76, 2.63] |
| Coral cover | 2009–2016 | n = 26; 29.98 (14.69) | n = 45; 32.68 (9.30) | t(36.8) = −0.84, 0.41 | −0.23 [−0.64, 0.18] |
| Rugosity | 2009–2016 | n = 26; 1.22 (0.07) | n = 45; 1.22 (0.08) | t(69) = −0.03, 0.98 | −0.01 [−0.41, 0.40] |
| Human gravity | 2014 | n = 3; 1940.33 (1538.97) | n = 2; 4471.5 (5609.48) | t(3) = −0.72, 0.53 | −0.65 [−2.16, 0.96] |
| Fishing pressure | 2015 | n = 3; 119 (98.88) | n = 2; 153.5 (21.92) | t(3) = −0.46, 0.68 | −0.42 [−1.92, 1.14] |

Values reported reflect summary statistics across coral reef sites with and without significant social–ecological (s-e) network closure effects, as well as results from a two-sided, two-sample t test of their mean difference and estimated effect sizes. Rugosity is a measure of structural complexity; human gravity is a measure of human impacts that accounts for human population size and reef accessibility[49]. Benthic data to calculate coral cover and rugosity was unavailable in site A, thus "With s-e closure" for these metrics report means from sites B and C. Supplementary Table 7 provides evidence that there is no meaningful bias introduced by the inclusion of site A in our other metrics, including our metrics of ecological condition. Satterthwaite's formula was used to approximate the degrees of freedom for the two-sample t test and effect size estimates of NPP and rugosity to account for unequal variance.
CI confidence interval, SST sea surface temperature, NPP net primary productivity

**Table 2 Key social processes**

| Attributes | Measurement | Site | | | | |
|---|---|---|---|---|---|---|
| | | A | B | C | D | E |
| Trust | Trust in fishers, reported on a scale of 1–5 (none, more distrust than trust, half/half, trust more than distrust, trust all); mean/SD—FS | 3.93 (0.96) | 3.95 (0.98) | 3.84 (0.95) | 4.09 (**1.11**)* | 3.63 (**1.19**)* |
| Common understanding/shared vision | Perception of resource state, where respondent reported there were less (−1), the same (0), or more (1) fish on reef than 5 years prior; mean/SD—FS | −0.82 (0.55) | −0.92 (0.35) | −0.84 (0.53) | −0.67 (**0.72**)* | −0.87 (0.44) |
| Commitments (rules) | | | | | | |
| Closed area | Yes/no—CL | Yes | Yes | Yes | Yes | Yes |
| Access rights | Yes/no—CL | Yes | Yes | Yes | Yes | Yes |
| Gear restrictions | Yes/no—CL | Yes | Yes | Yes | **No** | **No** |
| Conflict resolution mechanisms | Yes/no—CL | Yes | Yes | Yes | Yes | **No** |
| Internal conflicts | Reports of conflict within the community over gear use; yes/no—ref. [53] | No | No | No | **Yes** | **Yes** |

Indicators of key social processes theorized to be supported by social–ecological network closure across the five coral reef fishing communities studied (sites A–E). Notable differences are reported in bold. Asterisk (*) indicates a significantly different variance (p < 0.05) than those reported without an asterisk footnote according to Levene's robust test statistic for the equality of variances between groups.
FS fisher survey, CL community leader interview

(Table 3). All had developed rules adapted to the local condition, the ability to exclude outsiders, graduated sanctions, monitors that were locally accountable, and high levels of participation in decision-making (Table 3). Hence, none of these conditions could explain the observed differences in biomass and functional richness of fished resources.

## Discussion

Our quantitative and qualitative results provide evidence that closed social–ecological network structures among direct resource competitors may facilitate more effective cooperation that is associated with positive ecological conditions in coral reefs. In these multi-resource commons settings, the distinction between cooperation in a general sense and the more precise form of cooperation evaluated here that accounts for complex social–ecological interdependencies appears to be an important one. Indeed, results from our network models demonstrate that all study sites have a baseline propensity for cooperation among social actors (indicated by the significant, positive parameter estimates for social network closure, Fig. 3). This result supports recent research on the risk hypothesis[20], which argues that social actors tend to form closed, triadic social network structures to manage high-risk cooperation problems due to their ability to help develop and sustain trust and exert social pressure to comply with rules. Yet, despite this baseline tendency for cooperation across all sites, our results demonstrate that only sites A–C have a propensity for cooperation that results in social–ecological alignment by directly binding those who are dependent on the

same resources (social–ecological network closure, Fig. 3). Importantly, sites A–C also had higher levels of both biomass and functional richness of fished resources (Fig. 4), and these ecological conditions do not appear to be related to other network effects (Fig. 3); biophysical, environmental, or human impact characteristics (Table 1); or potentially confounding social and institutional factors (Table 3).

We proposed several theoretical mechanisms by which social–ecological network closure capturing cooperative communication among direct resource competitors might impact ecological conditions in this setting: i.e., the development trust, a shared vision, and the establishment of commitments among direct resource competitors toward sustainable resource management (Fig. 2). Our exploratory evaluation of these social processes was largely in line with our theoretical predictions. Specifically, we found that sites with a propensity for social–ecological network closure (sites A–C) demonstrated less variation in trust; a higher level of agreement on the state of reef resources; and a stronger commitment to sustainably managing reef resources, demonstrated by the establishment of a greater number of rules and avenues for conflict resolution (Table 2). This is important because reaching a consensus regarding what actions to take to manage common-pool resources such as reef fisheries and whether they will be effective is likely to be more difficult where there is less agreement about the state of the resource system and about whether people—especially direct resource competitors—can be trusted, e.g., to comply with devised rules[51]. Indeed, although our sites without a propensity

**Table 3 Social and institutional conditions**

| Attributes | Measurement | Site | | | | |
|---|---|---|---|---|---|---|
| | | A | B | C | D | E |
| Dependence on resource | Percentage of respondents who ranked fishing as their primary livelihood—FS | 92% | 85% | 92% | 70% | 99% |
| Rights to devise institution | Yes/no—ref. [53] | Yes | Yes | Yes | Yes | Yes |
| Ability to exclude outsiders | Yes/no—CL | Yes | Yes | Yes | Yes | Yes |
| Organizational experience/leadership | Yes/no—ref. [53] | Yes | Yes | Yes | Yes | Yes |
| Rules adapted to local condition | Yes/no—CL | Yes | Yes | Yes | Yes | Yes |
| Participation in decision making | Respondent was not (0), passively (1), or actively (2) involved in decisions about resource management, or held a leadership position (3); mean/SD—FS | 0.76 (0.73) | 0.98 (0.80) | 0.62 (0.72) | 0.68 (0.72) | 0.74 (0.67) |
| Monitors locally accountable | Yes/no—ref. [53] | Yes | Yes | Yes | Yes | Yes |
| Graduated sanctions | Yes/no—CL | Yes | Yes | Yes | Yes | Yes |

Indicators of social and institutional conditions known to be associated with effective collective management of the commons across the five coral reef fishing communities studied (sites A–E)
FS fisher survey, CL community leader interview

for social–ecological network closure (sites D and E) had devised some rules at the time of data collection, previous research[53] suggests that these rules were not easily established (e.g., McClanahan et al.[53] found that they experienced substantial delays in designating areas closed for fishing after indicating initial interest compared to other sites). Moreover, sites without a propensity for social–ecological network closure had not agreed on and instituted gear restrictions, which play a key role in managing reef fisheries because they modify fishing behavior rather than trying to prevent it[54]. This distinction is important because many reefs are located in developing countries, where more stringent regulations can undermine livelihoods and be difficult to enforce[54].

Practically, our results suggest that investments in building community capacity that specifically focus on establishing communication channels among direct resource competitors may improve reef ecosystem conditions. Yet given the competitive nature of many common-pool resource systems such as reef fisheries[55], important questions remain regarding how these relationships can be built. Here key social–ecological interactions were defined as those that linked fishers to specific species based on their fishing gear (Fig. 1). Our results thus suggest that stimulating gear-based communication may indirectly lead to a greater propensity for social–ecological network closure since the same set of species tend to be targeted by the same gear (Supplementary Table 3, Supplementary Methods). These communication channels can be facilitated by creating communities of practice centered around gear and technology, which can act to stimulate learning, build trust, and enhance shared ecological understanding of factors important for resources to be sustained[56]. However, caution is warranted, as efforts to build such communities of practice could lead to the emergence of competing gear-based coalitions and a zero-sum game where the potential ecological benefits from restricting one gear are captured by users of another gear[36]. This is a genuine risk in multi-species, multi-gear reef fisheries and other similar common-pool resource systems, where gear competition is ubiquitous. Thus broader community-building strategies that seek to establish communication and trust across all direct resource competitors, including actors using different gear types but overlapping in target species, is critical for achieving long-term sustainability. Notably, this communication may not need to be maintained over the long term, as recent research suggests that communication can have a persistent effect on cooperation in social dilemmas even after it has been removed[28]. What is critical, however, is that communication occurs long enough to establish prosocial norms that can activate guilt if and when someone considers defecting[28].

This study represents the first multi-site comparative analysis to examine how key aspects of social–ecological networks relate to quantitative ecosystem conditions. It therefore fills a critical gap in advancing integrative social–ecological network approaches for environmental problem-solving, which has been repeatedly advocated in recent years[6,7,9]. Applying this approach, we tested an important theoretical question regarding how social–ecological alignment relates to ecological conditions. Future research can extend this work to empirically test theory-driven hypotheses regarding other types of social–ecological interdependencies at various scales that may have important impacts on sustainability outcomes. For example, if coupled with dynamic or longitudinal data, this framework could be used to test explicit hypotheses about how changes in social structures drive the formation or dissolution of ecological links. The framework could also be used to explicitly capture social–ecological feedbacks, which have been difficult to study empirically.

Given the multitude and scale of anthropogenic drivers affecting the environment[32] and the costs associated with cooperation[55], understanding who should cooperate with whom in different contexts and to address different types of environmental problems is becoming increasingly important[14]. The benefit of the interdisciplinary social–ecological network approach described here is that it allows for a much more nuanced and precise understanding of the interdependencies between social and ecological components of ecosystems, allowing one to unpack the specific types of cooperative connections that facilitate or hinder effective action. Employing this approach, we provide evidence that social–ecological network closure—fully linked and thus closed, network structures between social actors and ecological resources—supports key social processes that may promote more effective collective management of shared resources, having demonstrable ecological impacts. Our results suggest that investments in building community capacity that focus on establishing communication, trust, and a shared understanding among direct resource competitors may improve ecological conditions in coral reef fisheries.

## Methods

**Summary of our empirical strategy**. We studied five coral reef fishing communities along the Kenyan coast. To test our hypothesis, we used a combination of quantitative and qualitative interdisciplinary data collected via semi-structured fisher surveys, underwater visual census, observed fish landings, key informant and expert interviews, and published reports[53]. Specifically, we drew on information from our fisher surveys, observed fish landings data, published reports, and expert interviews to construct full social–ecological networks akin to Fig. 1 for each study site. We then tested whether and to what extent the closed, cross-level social–ecological triangle (i.e., social–ecological network closure, Fig. 1) helped to explain the empirically observed structural characteristics of these networks using multilevel ERGMs. Next, we tested for differences in ecological resource conditions within fished areas of sites with and without social–ecological network closure using underwater visual census data. We also tested for differences in key biophysical, environmental, and human impact characteristics known to affect reef ecosystem conditions. We then drew on information from our fisher surveys, conducted key informant interviews, and reviewed published reports to explore whether the key social processes we argue are supported by social–ecological network closure were present in each site (i.e., Fig. 2). We also used this information to assess whether other social and institutional conditions associated with effective management of the commons[25,51] may have affected ecological resource conditions across sites.

**Site selection**. Sites were selected from a ~100 km stretch of the Kenyan coast (Supplementary Fig. 1) in collaboration with our partners at the Wildlife Conservation Society's Coral Reef Conservation Program (TRM). We specifically chose sites (1) that were relatively close together to minimize differences in key biophysical and environmental conditions, (2) where fishing was the primary occupation of the majority of the population, (3) where our partners had been engaged in monitoring, and (4) where communities were considered to have achieved a range of success in managing reef fisheries resources collectively as a community in order to combat declining trends (Supplementary Methods). Each site selected was comprised of a social community of fishers and an associated fishing area adjacent to their community that they use and have rights to manage (see Supplementary Information for more details). All fishing areas sampled were shallow (<10 m depth), exposed to similar environmental conditions (Table 1), and have a similar disturbance history (e.g., coral bleaching).

**Constructing the social–ecological networks**. To capture cooperative communication relationships among fishers (i.e., the social network A, Fig. 1), we administered a semi-structured fisher survey from December 2015 to May 2016. A total of 711 fishers were originally surveyed, representing 75–84% of the total estimated population of fishers within each site (Supplementary Table 1). Eighty-one fishers were subsequently dropped owing to missing information (Supplementary Table 1). We used a name generator with qualifiers (Supplementary Methods), where fishers were specifically asked to nominate up to ten individuals with whom they exchanged information and advice with about fishing and fishery management (e.g., rules, gears, and fishing locations). Name qualifiers were checked daily with local guides while fieldwork was being conducted to ensure identification accuracy of all nominated individuals. Non-respondent network actors were dropped and ties were symmetrized and treated as binary. The corresponding social networks were thus undirected, with edges representing information and advice relationships between respondents $A_i$ and $A_j$ in each site (Supplementary Table 1, Supplementary Fig. 1). Fishers were also asked to report what type of fishing gear they used in addition to other sociodemographic characteristics that existing research suggests plays a role in structuring social interactions in fisheries, e.g., ethnicity, leadership, and landing site[46] (Supplementary Table 2). Surveys were conducted via in-person interviews in Swahili.

The ecological network (B, Fig. 1) captures trophic interactions among target fish species comprising the majority of catch by all fishing gears employed in our five study sites ($n = 36$ species, Supplementary Methods; Supplementary Fig. 2). Target fish species for each gear type were identified using detailed landings data from 25 landing sites along the Kenyan coast collected continuously between 2010 and 2016 (Supplementary Table 3). Trophic interactions (i.e., predator–prey relationships) were estimated based on a combination of diet, relative body size, and habitat use[18,57,58] (Supplementary Methods). The corresponding ecological network was thus undirected, with edges representing trophic interactions between fish species $B_u$ and $B_v$. Social–ecological ties ($X$, Fig. 1) were identified by linking individual fish species to individual fishers via their primary fishing gear as identified in the fisher survey (Supplementary Table 4). In other words, if fisher $A_i$ used gear type $G_t$ as their primary gear, and gear type $G_t$ targeted fish species $B_u$, a social–ecological link would exist between fisher $A_i$ and fish species $B_u$.

**Multilevel network models**. We used multilevel exponential graph models (ERGMs) (Supplementary Methods) to test the prevalence of the closed, cross-level social–ecological triangle configuration representing cooperative communication among direct resource competitors within each site. ERGMs are statistical models of networks based on explicit hypotheses about network dependence[59]. ERGMs model network ties explicitly by treating each tie as a random variable and

specifying the probability of observing the network ($Y$) with $n$ nodes as a function of various local network processes. These network processes are expressed as micro-level network configurations (e.g., edges, stars, and triangles) where all ties are assumed conditionally dependent. The dependence assumption is key because it captures the idea that, rather than forming at random, empirical network ties self-organize into various patterns arising from underlying social processes[60], e.g., preferential attachment[45] and transitivity[19]. The observed network structure is thus seen as one possible outcome of these stochastic network processes. Multilevel ERGMs can be seen as an extension of ERGMs that account for networks linked across multiple levels[44]. Here, network ties are considered interdependent not only within levels but also across levels, enabling the interpretation of cross-level interactions and configurations (e.g., Fig. 1). In this study, we employed an extended version of multilevel ERGMs that builds on social selection models[61] to incorporate nodal attributes as exogenous covariates in order to account for their ability to effect network structures (Supplementary Methods).

We tested for social–ecological network closure—i.e., the closed, cross-level social–ecological triangle depicted in Fig. 1—while controlling for nodal attributes known to shape social interactions among fishers and other well-known mechanisms involved in shaping social networks[59]. Nodal attributes included were (1) leader activity (the propensity for leaders to be active/have more ties in the network) and (2) landing site homophily (homophily among fishers using the same landing site), as these have been shown to affect social tie formation in small-scale fisheries[46]. Full models also included controls for activity in each landing site where a residual analysis[62] suggested that fishers associated with that landing site were more active in forming and maintaining ties than would be expected by chance alone (Supplementary Table 5). To control for endogenous mechanisms in the social network, we included (1) the edge parameter to capture density, which corresponds to the baseline propensity to establish ties; (2) centralization parameters (the alternating star and a two-star parameter where appropriate; Supplementary Methods) to capture preferential attachment; and (3) the alternating triangle parameter to capture transitive closure.[2]

Because the focus here was on social processes, and particularly the propensity for fishers to form ties with direct resource competitors, the $X$ and $B$ level networks (Fig. 1) were fixed and treated as exogenous; in other words, their structure was treated as given and therefore ties within these levels were not explicitly modeled. Goodness-of-fit tests and residual analyses demonstrated that nearly all graph characteristics were well accounted for by our final models (Supplementary Methods, Supplementary Table 6). Note that, in multi-level ERGMs, the parameter estimates for cross-level effects (e.g., social–ecological network closure) cannot be directly compared to the parameter estimates for within-level effects (e.g., social network density). Mahalanobis distances for each model indicated a better model fit with the inclusion of the cross-level social–ecological triangle (Supplementary Methods). All models were run in MPNet[63], which implements a Markov Chain Monte Carlo procedure to estimate model parameters using maximum likelihood estimation[64].

**Assessment of ecological conditions**. We used detailed underwater visual census data collected between 2010 and 2015 that surveyed fish in replicate 500 m$^2$ transects at each site (Supplementary Methods, Supplementary Table 7) to generate our estimates of biomass and functional richness of fished resources. Using this data, we tested for mean differences in reef fish biomass and functional richness between sites with and without social–ecological network closure using a one-sided, two-sample $t$ test and effect size estimates (Cohen's D). We conducted identical tests on all available data (2010–2015) and on data from 2014 only (which most closely matches when our social data was collected) and found no difference in our results (Supplementary Table 8). Satterthwaite's formula[65] was used to approximate the degrees of freedom for all tests where the data was found to have unequal variance across groups.

**Identifying key social processes**. To explore the presence of, and variation in key social processes theorized to be supported by social–ecological network closure (Fig. 2, Table 2), we drew on our fisher survey, community leader interviews, and existing research[53]. Specifically, we examined trust using a five point Likert-scale variable in our fisher survey, where fishers were asked to report how much they trusted other fishers. To assess whether fishers had a common understanding or shared image, we asked how they perceived the state of the resource system in our fisher survey (i.e., was there more, the same, or less fish on the reef than 5 years ago?). We compared the variation in trust and fisher's perceptions of the state of the resource system across sites using Levene's test for the equality of variance. We compared mean levels of trust using Mann–Whitney $U$ non-parametric test. The results of this test were insensitive to whether one applied this procedure; a two-sided $t$ test accounting for unequal variance using Satterthwaite's approximation for degrees of freedom; or a linear mixed model with a fixed effect for group (i.e., sites with and without significant social–ecological network closure effects) and a random effect for individual, which accounts for the non-independent nature of observations from the 45 fishers (out of 648) who identified themselves as part of two of our study communities. To assess the level of commitments made within each site regarding the management of fishery resources, we interviewed community leaders to examine the rules in use and whether conflict resolution mechanisms had been established. Reports of within-community conflict were described in the existing research[53].

**Accounting for potentially confounding factors**. We assessed differences in key biophysical, environmental, and human impact characteristics known to effect reef ecosystem condition between sites with and without social–ecological network closure using a two-sided, two-sample $t$ test and effect size estimates (Cohen's $D$; Table 1). Satterthwaite's formula[65] was used to approximate the degrees of freedom for all tests where the data was found to have unequal variance across groups. Biophysical variables were hard coral cover[66] and rugosity, a measure of the structural complexity of the habitat[48]. Environmental variables were SST and NPP. Human impact measures were fishing pressure and human gravity[49], a metric that accounts for human population and reef accessibility (including travel time[50]) that aims to capture both market and subsistence pressures on reefs. Data sources are further detailed in Supplementary Table 7. To assess relevant social and institutional conditions within each site (Table 3), we examined the prevalence of, and variation in Ostrom's[25] institutional design principles shown to support robust management of the commons[52]. Specifically, we interviewed community leaders to determine whether each site had the ability to exclude outsiders, if rules were adapted to local conditions, whether graduated sanctions were in place, and if conflict resolution mechanisms existed. We drew on existing research[41] to determine whether monitors were locally accountable and whether communities had rights to devise their own institutions without being challenged by external governing authorities. We used our fisher survey to assess mean levels of participation in decision making about resource management issues using two approaches: a two-sided $t$ test for equality of means, and a linear mixed model with a fixed effect for group (i.e., sites with and without significant social–ecological network closure effects) and a random effect for individual, which accounts for the non-independent nature of observations from the 45 fishers (out of 648) who identified themselves as part of two of our study communities. The results were insensitive to the approach used. Using information from our fisher survey and published reports[41], we also examined two attributes known to be positively related to collective action in the commons: (1) salience, i.e., the majority of resource users are dependent on the resource system to support their livelihoods, and (2) prior organizational experience and local leadership[51].

**Limitations**. Common to empirical inquiries attempting to uncover network effects[67], our comparative analysis is not without limitations. First, owing to the high data demands of our approach and the intensive nature of collecting detailed and complete, empirical social networks, we were only able to study five communities. Despite this, the results of our multilevel ERGMs and ecological conditions provide support for our hypothesis, and we were able to further support our inferences by incorporating a range of additional data characterizing key social processes; biophysical, environmental, and human impact characteristics; as well as the social and institutional conditions in each community. Second, because we collected detailed social network data in addition to data on fishing behaviors and other social factors, the amount of time spent on each topic in our interviews had to be carefully considered in order to avoid respondent fatigue. Thus we were only able to gain preliminary empirical insights into the mechanisms by which social–ecological network closure can affect ecological conditions (Fig. 2). Mechanisms—particularly those that involve human behavior—are difficult to isolate and study empirically in field settings. As an example, we assessed variation in perceptions over the state of reef resources to gauge whether fishers had a common understanding or shared image of the resource system and how it operates (Table 2); yet it is possible that variation in fisher's perceptions of the state of the resource could potentially be due to more complex or less obvious resource dynamics. Still, the mechanisms proposed here have strong theoretical support[27,28,30]. Our empirical assessment of these social processes should thus be seen as exploratory in nature, and only one part of a triangulation effort to more thoroughly test our claims linking social–ecological network closure to ecological conditions. Third, our approach relied on cross-sectional network and socio-economic data, preventing us from establishing clear temporal trends and causality between social–ecological network closure and ecological conditions. This is a common limitation in empirical social–ecological research due to high data demands and is particularly pronounced with empirical network research. However, our inquiry was grounded in well-established theories of communication and cooperation, giving us a high level of confidence that our results point to social–ecological network closure as a predecessor to improved ecological conditions, rather than the reverse. More firmly establishing casual links would require integrative, interdisciplinary social and ecological data collected at multiple points in time—a task likely to require a career of work but could be more efficiently facilitated by long-term collaborative endeavors.

**Ethics statement**. Research protocols were approved by the Institutional Review Board of the Office of Research Compliance Human Studies Program at the University of Hawaii at Manoa and the Human Ethics Research Committee at James Cook University. Informed consent was obtained from all respondents.

**Reporting summary**. Further information on research design is available in the Nature Research Reporting Summary linked to this article.

## Data availability

Summary ecological and social data that support the findings of this study are available within the paper and its Supplementary Information files. Raw ecological network data have been deposited in the Tropical Data Hub and can be accessed at https://doi.org/10.25903/5c89d99f5d654. Raw social, social network, and social–ecological network data are available upon request from the corresponding author M.L.B. with reasonable restrictions, as these data contain information that could compromise research participant privacy and consent.

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

## Acknowledgements

This project was supported by an NSF Social, Behavioral, and Economic Sciences Postdoctoral Research Fellowship Grant to M.L.B. (#1513354), the ARC Centre of Excellence for Coral Reef Studies, the Darwin Initiative (#20–017), and the Marine Science for Management Contract No. MASMA/OP/2014/04. Ö.B. was supported by the Swedish Foundation for Strategic Environmental Research (Mistra) through a core grant to the Stockholm Resilience Centre at Stockholm University. We thank G. Robins, G. Cumming, J. Cinner, and our reviewers for helpful comments on earlier drafts, the Wildlife Conservation Society's Coral Reef Conservation Program for data access and logistical support, Stephen Wanyonyi and Innocent Muly for their assistance in the field, Emmanuel Mbaru for help with data processing and graphics, and all of the fishers who participated in this project.

## Author contributions

M.L.B. designed the integrated social–ecological research. T.R.M. designed the research on ecological conditions and biophysical characteristics. M.L.B., T.R.M., A.S.H., and N.A.J.G. performed the research. M.L.B., Ö.B., and T.R.M. analyzed data. M.L.B., Ö.B., T.R.M., J.N.K., A.S.H., O.G.G., and N.A.J.G. wrote the paper.
