## [Peer Review File · Nature Communications]

Reviewers' Comments:

Reviewer #1:

Remarks to the Author:

This paper investigates the important topic of social-ecological linkages in coral-reef fisheries. While these linkages are clearly critical, there are few quantitative analyses beyond correlating fishing and fish assemblages. This paper is a significant addition to the literature by quantifying the detailed structure of social-ecological networks and showing the characteristics that seem to result in higher fish biomasses and functional diversities. These characteristics also appear to be more informative than a proxy of fishing pressure (fishers / km²). I found the paper insightful, well written, and the data appear well analyzed using cutting edge techniques – although I am not familiar enough with the analytical methods to assess their implementation in detail.

My only substantial comment is that I feel like the fish data need better explanation. Firstly, it was strange that there was no map of sites, so readers have no idea of the scale of site separation. This is useful to know, but also is important for interpreting the fish results – which are critical because they provide the validation for the network analyses revealing characteristics that seem to benefit fish populations. Spatial scale here is important because there is a growing recognition that maximum biomass on reefs varies spatially depending on temperature, productivity etc. Readers need to be assured that the % of pristine biomass in Fig 3A are relatively robust – showing that the sites were relatively close together would do this (limited biophysical gradients) or otherwise showing the key gradients (e.g. SST, NPP) do not vary systematically from A – E. Furthermore, there is no information that I could see on whether surveys were done in similar habitats – again readers need to be assured there are no systematic biases from A – E as it is well established that varying habitat type can change fish assemblages dramatically. Do the authors have information on depth / habitat type / coral cover / rugosity?

Reviewer #2:

Remarks to the Author:

Summary:

This paper tackles a challenging and important topic, the role of cooperative communication relationships among resource users in driving positive ecological outcomes. The authors used a social-ecological network approach, which appears to be an elegant way to test theory-driven hypotheses regarding key social-ecological interactions in multi-resource commons settings. The paper has several important results, suggesting that bonding social capital (operationalized through social-ecological network closure) can lead to improved ecological conditions in the commons. The analysis presented identifies mechanisms by which such social interactions can lead to positive outcomes (improved trust, shared vision, common understanding and commitment to sustainably managing resources). These findings are then linked with management interventions, highlighting potential for stimulating gear-based communication to improve on sustainability.

Review:

While the paper is well-written, and tackles an important topic, it does not seem suitable for publication in Nature Communications in its current form. There is much that is good about this paper - it provides one avenue to operationalize key social-ecological concepts, but the paper has two related flaws in the argument: i) an overly simplistic estimate of 'ecological outcome', and ii) an inappropriate study design to infer causal links between social-ecological network closure and ecological outcome. Discussion of each of these main points follows. Social-ecological fit and sustainability is an important topic, but this paper will have to address these issues in a more detailed

and complete manner. I do not have experience with multilevel exponential random graph models (ERGMs) so have not been able to comment on the appropriateness of some of the methods used.

(1) Issues surrounding the 'ecological outcome'

An important aspect of the paper is the estimate of the 'ecological outcome', and especially the relationship between social-ecological network closure and fish biomass and functional richness at each site. Authors show that sites where fishers successfully formed cooperative communication ties have higher % fishable biomass and functional richness, and claim that this "provide strong evidence that when fishers facing commons dilemmas form cooperative communication ties with direct resource competitors, they can achieve positive gains [in ecological outcomes]". However, the 'ecological outcomes' as calculated and analyzed in this submission do not seem rigorous.

First, the role of socioeconomic, biophysical and governance factors in shaping fish communities is complex and varies through space and time. Although authors investigate the potential effect of fishing pressure, and other confounding factors related to Ostrom's design principles, they completely ignore other well-known features (e.g., depth, habitat, age of the organization, history of disturbance, proximity to other human disturbances, etc.) that could explain the differences observed among sites. The low number of sites (n=5) and the apparently local nature (information on site location are missing) of this study exacerbate this issue and casts doubts about the robustness of the conclusions.

Fig. 3 shows fish biomass and functional richness across study sites. This essentially reflects spatial variability. Biomass in sites B-C, and functional richness in site C are indeed higher than in D-E, but the difference is small. The absence of confidence interval does not allow to appreciate the variability among sites (confidence intervals are only provided for species richness in D-E, why?). Given the importance of this part of the analysis in authors' conclusion, they should at least provide a more rigorous analysis of among-sites difference by choosing an appropriate statistical test of significance (hopefully showing that D-E and significantly different from A-C). A qualitative judgment solely comparing the two extreme cases (A and E) seems clearly insufficient. In addition, surveying the fish community along 1-2 500m² transects at only one point in time seems rather small to accurately depict the state of the fish community at each site. Most studies I am familiar with in the international literature, from both tropical and temperate systems, would use greater spatial and temporal replication to reduce sampling variability and increase statistical power.

Also, no information is provided on the spatial features of sites/fishing communities. Given that in some cases, fishers from different communities share the same landing sites, it is safe to assume that they are geographically close to each other. I wonder whether their fishing ground could overlap in some cases, in which case it may be even more challenging to differentiate the state of the resource between sites?

More importantly, I don't think the term 'ecological outcome' actually reflects the meaning of the metrics used. Indeed, fish absolute biomass and functional richness reflect (part of) the state of the resource, not the outcome of social interactions. The term 'resource unit' (sensu Ostrom 2009) might be more defensible, and less misleading. Unfortunately, the measure of 'ecological outcome' is so central to the results and discussion presented in the paper that it seems unwarranted to accept the work as presented. Authors should either reformulate their hypothesis to explicitly acknowledge this point (i.e., they test if social-ecological network closure is associated with positive ecological state (and not outcome) in the face of the commons dilemma), or use an appropriate measure of 'outcome' to support their claim. In the first option, proceeding with the rest of the analysis would potentially imply accounting for different social and ecological processes (and feedbacks?) that the ones present in the current paper. The second option may be achieved through Before-After (BA), Control-Impact (CI), or ideally BACI design (Bence et al. 1996, Stewart-Oaten & Bence 2001).

Bence, J.R., Stewart-Oaten, A., Schroeter, S.C., 1996. Detecting ecological impacts, in *Detecting Ecological Impacts: Concepts and Applications in Coastal Habitats*

Ostrom, E., 2009. A general framework for analyzing sustainability of social-ecological systems. *Science* 325, 419–22.

Stewart-Oaten, A., Bence, J.R., 2001. Temporal and spatial variation in environment impact assessment. *Ecol. Monogr.* 71, 305–339.

(2) Inference about potential mechanisms by which social-ecological network closure impacts 'ecological outcomes'

To support their claim that social-ecological network closure enhances ecological outcomes, authors incorporate additional quantitative and qualitative data characterizing key processes of social-ecological systems. They conclude that the development of trust, shared vision, as well as commitments toward sustainable resource use are the main mechanisms involved in their case.

Beyond the fact that they did not include key biophysical processes (see point above), I am not convinced by the approach. Trust, for instance, is shown as being significantly less variable in sites with highest propensity for social-ecological network closure. Although measuring variation in trust is interesting, I was surprised to find in Table 2 that Trust was, however, highest in D, a site with low propensity for social-ecological network closure. Among-sites differences in mean seems (in the case of trust) more appropriate, and authors should justify why they chose to look only at variability instead.

Similarly, although I found the focus on among-sites differences in variability more appropriate in the case of 'common understanding/shared vision', the metric used is questionable. Indeed, authors interpret high variation in fisher's perceptions of the state of the resource as a lack of similar understanding of how a resource works and what rules and norms are shared by others. However, one might argue that high variation in such metric could also be due to more complex, or less obvious resource dynamics. Values provided in Table 2 suggest that this might be the case, since the only site with significantly higher variability (site D) is also the one where respondents reported lower decreases in the resource.

I would be interested to know what makes site A stand out so much from the others when looking at the high biomass/functional richness and high fishing pressure. Authors reject the idea of a potential contribution of fishing pressure to among-sites differences in 'ecological outcome' solely based on this outlier, but do not mention that fish biomass and functional richness are negatively related to fishing pressure.

In the Discussion, authors claim that "the benefit of the interdisciplinary social-ecological network approach described here is that it allows for a much more nuanced and precise understanding of the interdependencies between social and ecological components of ecosystems, allowing one to unpack the specific types of cooperative connections that facilitate or hinder effective action.". Unfortunately, the examples above highlight that despite the great level of details in the social-ecological network analysis used in this study, serious methodological concerns remain and a tendency to emphasize particular cases to support hypothesized mechanisms undermine an otherwise interesting and original study.

In summary, my view is that despite some exciting methodological and operational contributions (i.e., social-ecological network approaches applied to multi-resource commons), the main claim of this submission (that “closed social-ecological network structures amongst direct resource competitors facilitates more effective cooperation that can promote positive ecological outcomes in coral reefs”) is undermined by inferences that are cursory at best (due to analyses that are not especially rigorous or thorough), and hence unfortunately provides conclusions that are not particularly robust.

Other comments:

The authors have done a very good job of limiting jargon, and presenting their results clearly. A little more detail in the text and/or in SI (in particular, general characteristics of each sites beyond those provided in SI; these may include location, governance arrangements, etc.) would help the reader evaluate fairly the research presented.

I found causal pathway described in this study overly simplistic. I am surprised that feedbacks from the ecological to the social systems (yet present in the SES framework this paper seems to rely on for the theoretical mechanisms) were not considered.

I don't understand how and when the ecological network (representing trophic interactions among species) was used in this study. Please clarify its contribution to the study findings.

Line 10: According to Table S1, 63 fishers were dropped from the analysis so the number here should be 693 instead of 756.

Reviewer #3:

Remarks to the Author:

This paper seeks to understand the relationship between multi-level network properties in socio-ecological systems and its relationship to ecosystem viability using ERGMs. This is an interesting approach that directly links natural elements of ecosystems with human behavior. The approach is novel but there needs to be more background information to help in interpreting the statistical findings of the study. It would be nice to know the spatial distribution of the reef systems and the spatial and community composition of the fishers (e.g., maps might help). To get a better understanding of the relationship between landing sites and reefs a 2-mode matrix involving reefs and landing sites was created from data in the supplemental materials. Basically, based on the ecological assessments sites A, B and C appear to be more ecologically viable while reefs D and E are less so. Correspondingly, there is multi-level network closure in the networks of the viable systems but not in the other two reefs. The attached graph is the two mode network of reefs by landing sites. What is interesting is that reef E, one of the less viable reefs, clearly uses very different landing sites. However, reefs A and D are essentially structurally equivalent in the graph yet one reef system is more viable than the other. It appears that the fishers at the two reefs use the same landing sites, yet have separate networks. Further background information may help in better clarifying this observation. The question however, is given the similar behaviors in choices of landing sites between the fishers at the two reefs, what kinds of interactions occur between the fishers from the different reefs? Without this explanation it is difficult to interpret the study's findings.

Also, fishers around the world tend to engage in targeting multiple species with multiple gears. Do fishers in this context only use one type of gear when targeting species or do they engage in switching behaviors throughout the year? This needs clarification.

● 3

● 11

Reviewer 1	Author Response
This paper investigates the important topic of social-ecological linkages in coral-reef fisheries. While these linkages are clearly crucial, there are few quantitative analyses beyond correlating fishing and fish assemblages. This paper is a significant addition to the literature by quantifying the detailed structure of social-ecological networks and showing the characteristics that seem to result in higher fish biomasses and functional diversities. These characteristics also appear to be more informative than a proxy of fishing pressure (fishers / km²). I found the paper insightful, well written, and the data appear well analyzed using cutting edge techniques – although I am not familiar enough with the analytical methods to assess their implementation in detail.	Thank you for the positive comments. We agree that social-ecological linkages are important for driving outcomes in coral reef fisheries, and believe that the network approach we demonstrate here has potential to help us untangle some of these key relationships.
My only substantial comment is that I feel like the fish data need better explanation. Firstly, it was strange that there was no map of sites, so readers have no idea of the scale of site separation. This is useful to know, but also is important for interpreting the fish results – which are crucial because they provide the validation for the network analyses revealing characteristics that seem to benefit fish populations. Spatial scale here is important because there is a growing recognition that maximum biomass on reefs varies spatially depending on temperature, productivity etc. Readers need to be assured that the % of pristine biomass in Fig 3A are relatively robust – showing that the sites were relatively close together would do this (limited biophysical gradients) or otherwise showing the key gradients (e.g. SST, NPP) do not vary systematically from A – E. Furthermore, there is no information that I could see on whether surveys were done in similar habitats – again readers need to be assured there are no systematic biases from A – E as it is well established that varying habitat type can change fish assemblages dramatically. Do the authors have information on depth / habitat type / coral cover / rugosity?	Thank you for this comment. We have now included additional context, data, and analyses to further support our inferences and we believe these additions have made our contribution stronger. Specifically, we now:  1. Include a map of sites (Fig. S1), which shows that all sites are nearshore, positioned along a ~100 km stretch of the Kenyan coastline, and therefore likely to experience similar environmental conditions. 2. Incorporate all available data from our sites on key biophysical and environmental variables (i.e., SST, NPP, coral cover, and rugosity) as well as human impact measures (human gravity following Cinner et al. 2018 PNAS, and fishing pressure); and present summary data, two-sample t-tests, and effect size estimates in Table 2 that demonstrate there is no substantial variation among these variables in sites with and without social-ecological network closure. Table S7 summarizes these data, data sources, and methods. 3. Include text in the methods section that describes the habitat of each site, which reads: “All fishing areas sampled were shallow (<10m depth), exposed to similar environmental conditions (Table 2) and have a similar disturbance history (e.g., coral bleaching).” In response to Reviewer 2, to further support our inference that social-ecological network closure supports key ecological conditions in coral reefs, we also now include a two-sample t-test and effect size estimates for biomass and functional richness between sites with social-ecological network closure and those without (see Fig. 3, and ‘Ecological conditions’). The results of these tests suggest that the mean differences between sites with and without social-ecological network closure are significant and the effect size is large. Because coral cover and rugosity data were unavailable for site A, we also ran a sensitivity analysis

	by running these same tests with site A removed (see Table S8 in the SI). Our results were unchanged, indicating that there is no meaningful bias introduced by the inclusion of site A.
--	--

Reviewer 2	Author Response
Reviewer 2 argued that this was an important topic, the paper was well-written, and that there was much that was good about this paper. They then highlighted two main points: (1) issues surrounding the ecological outcome, and (2) inferences about potential mechanisms by which social-ecological network closure impacts ecological outcomes. We begin our response with point 1 before moving on to point 2 below.	
(1) Issues surrounding the ‘ecological outcome’ An important aspect of the paper is the estimate of the ‘ecological outcome’, and especially the relationship between social-ecological network closure and fish biomass and functional richness at each site. Authors show that sites where fishers successfully formed cooperative communication ties have higher % fishable biomass and functional richness, and claim that this “provide strong evidence that when fishers facing commons dilemmas form cooperative communication ties with direct resource competitors, they can achieve positive gains [in ecological outcomes]”. However, the ‘ecological outcomes’ as calculated and analyzed in this submission do not seem rigorous. First, the role of socioeconomic, biophysical and governance factors in shaping fish communities is complex and varies through space and time. Although authors investigate the potential effect of fishing pressure, and other confounding factors related to Ostrom’s design principles, they completely ignore other well-known features (e.g., depth, habitat, age of the organization, history of disturbance, proximity to other human disturbances, etc.) that could explain the differences observed among sites. The low number of sites (n=5) and the apparently local nature (information on site location are missing) of this study exacerbate this issue and casts doubts about the robustness of the conclusions. Fig. 3 shows fish biomass and functional richness across study sites. This essentially reflects spatial variability. Biomass in sites B-C, and functional richness in site C are indeed higher than in D-E, but the difference is small. The absence of confidence interval does not allow to appreciate the variability among sites (confidence intervals are only provided for species richness in D-E, why?). Given the importance of this part of the analysis in authors’ conclusion, they should at least provide a more rigorous analysis of among-sites difference by choosing an appropriate statistical test of significance (hopefully showing that D-E and significantly different from A-C). A qualitative judgment solely comparing the two extreme cases (A and E)	We very much appreciated this comment, which played a major role in helping us to strengthen our inferences and arguments. We have made several adjustments to our analyses and the text in order to address this comment. Specifically, we now:  1. Include a map of sites (Fig. S1) as detailed above. 2. Incorporate all available data on biomass and functional richness of fished resources from 2010-2015 across sites, which increases our total sample from 9 to 18 individual transects and ensures there is replication in every site. As such, we now report confidence intervals for functional richness in all sites in Fig. 3. 3. Include more detail to justify our use of reef fish biomass and functional richness as metrics of ecological condition. Specifically, at the end of our introduction, we discuss how reef fish are key elements of the ecosystem that drive processes linked to ecosystem condition and stability (Graham et al. 2015 Nature). Fish biomass has been shown to capture a wide range of information on reef fish functioning (e.g. herbivory, predation), trophic structure, life history composition, and benthic ecosystem state (e.g. McClanahan et al. 2011 PNAS, 2015 Conserv Biol). We also discuss how functional richness is fast becoming a much preferred measure of biodiversity in ecology over species richness, as it captures more about the role of those species in ecosystem functioning (McGill et al 2006 Trends in Ecology & Evol; Mouillot et al. 2013 Trends in Ecology & Evol). 4. Include relevant statistical tests to show that mean levels of both biomass and functional richness are indeed statistically higher in sites with social-ecological network closure. We also present effect size estimates which indicate that these differences are large and meaningful (see Fig. 3, and ‘Ecological conditions’). 5. Include a sensitivity analysis (Table S8) demonstrating there is no meaningful difference in our results when

seems clearly insufficient. In addition, surveying the fish community along 1-2 500m² transects at only one point in Ume seems rather small to accurately depict the state of the fish community at each site. Most studies I am familiar with in the international literature, from both tropical and temperate systems, would use greater spatial and temporal replication to reduce sampling variability and increase statistical power.

Also, no information is provided on the spatial features of sites/fishing communities. Given that in some cases, fishers from different communities share the same landing sites, it is safe to assume that they are geographically close to each other. I wonder whether their fishing ground could overlap in some cases, in which case it may be even more challenging to differentiate the state of the resource between sites?

More importantly, I don't think the term 'ecological outcome' actually reflects the meaning of the metrics used. Indeed, fish absolute biomass and functional richness reflect (part of) the state of the resource, not the outcome of social interactions. The term 'resource unit' (sensu Ostrom 2009) might be more defensible, and less misleading. Unfortunately, the measure of 'ecological outcome' is so central to the results and discussion presented in the paper that it seems unwarranted to accept the work as presented. Authors should either reformulate their hypothesis to explicitly acknowledge this point (i.e., they test if social-ecological network closure is associated with positive ecological state (and not outcome) in the face of the commons dilemma), or use an appropriate measure of 'outcome' to support their claim. In the first option, proceeding with the rest of the analysis would potentially imply accounting for different social and ecological processes (and feedbacks?) that the ones present in the current paper. The second option may be achieved through Before-After (BA), Control-Impact (CI), or ideally BACI design (Bence et al. 1996, Stewart-Oaten & Bence 2001).

you restrict the sample to only data from 2014 versus the larger (2010-2015) dataset.

6. Include text in the methods section that describes the habitat of each site, which reads: "All fishing areas sampled were shallow (<10m depth), exposed to similar environmental conditions (Table 2) and have a similar disturbance history (e.g., coral bleaching)."

7. Incorporate all available data from our sites on key biophysical, environmental, and human impact variables that may influence reef ecosystem conditions (i.e., SST, NPP, coral cover, rugosity, gravity, fishing pressure), and present summary data, two-sample t-tests, and effect size estimates in Table 2 that demonstrate there is no substantial variation among these variables in sites with and without social-ecological network closure.

8. Because coral cover and rugosity data were unavailable for site A, we also performed a sensitivity analysis by running all tests described above with site A removed versus site A included (see Table S8). Our results were insensitive to the removal of site A, indicating that there is no meaningful bias introduced by the inclusion of site A.

9. Include text in the methods section under 'Site selection' and the beginning of the SI explaining more about the spatial features of the communities. Specifically, we explain that each community has its own associated fishing grounds. The only overlap was among a minority of fishers (n=45 out of 648) from sites A and D. These fishers fished in both sites A and D and considered themselves part of both fishing communities. Our qualitative insights from the on-the-ground data gathering efforts indicated that these fishers adjust their social behaviours depending on the context (i.e., which site they are operating/fishing in), which is consistent with research in social psychology (e.g., Turner et al. 1994 Personality & Social Psych Bulletin). We therefore accounted for this directly in our original analysis by including them in both networks - in other words, because they fish in both sites and therefore can have an effect on our measures of ecological conditions in both sites, whether or not they form cooperative communication ties with others from each site and target different species in each site is important, and was therefore considered in our original models. Nonetheless, during our revision process we re-ran our network models for sites A and D again with these fishers removed (as a sensitivity analysis, which we now describe in the SI), and found no meaningful difference in our results. This shows that these multi-site fishers alone are not significantly different from the majority of the other fishers only operating in one of the communities/fishing grounds.

	10. We see where the confusion may have arisen in reference to our usage of ‘ecological outcomes’. To add clarity, we have removed the reference to “ecological outcomes” and instead focus on “ecological conditions”. As discussed above (#3), we now provide much more detail to justify our use of fish biomass and functional richness as metrics of ecological condition.
(2) Inference about potential mechanisms by which social-ecological network closure impacts ‘ecological outcomes’ To support their claim that social-ecological network closure enhances ecological outcomes, authors incorporate additional quantitative and qualitative data characterizing key processes of social-ecological systems. They conclude that the development of trust, shared vision, as well as commitments toward sustainable resource use are the main mechanisms involved in their case. Beyond the fact that they did not include key biophysical processes (see point above), I am not convinced by the approach. Trust, for instance, is shown as being significantly less variable in sites with highest propensity for social-ecological network closure. Although measuring variation in trust is interesting, I was surprised to find in Table 2 that Trust was, however, highest in D, a site with low propensity for social-ecological network closure. Among-sites differences in mean seems (in the case of trust) more appropriate, and authors should justify why they chose to look only at variability instead. Similarly, although I found the focus on among-sites differences in variability more appropriate in the case of ‘common understanding/shared vision’, the metric used is questionable. Indeed, authors interpret high variation in fisher’s perceptions of the state of the resource as a lack of similar understanding of how a resource works and what rules and norms are shared by others. However, one might argue that high variation in such metric could also be due to more complex, or less obvious resource dynamics. Values provided in Table 2 suggest that this might be the case, since the only site with significantly higher variability (site D) is also the one where respondents reported lower decreases in the resource. I would be interested to know what makes site A stand out so much from the others when looking at the high biomass/functional richness and high fishing pressure. Authors reject the idea of a potential contribution of fishing pressure to among-sites differences in ‘ecological outcome’ solely based on this outlier, but do not mention that fish biomass and functional richness are negatively related to fishing pressure.	In addition to including a range of additional biophysical, environmental, and social data (discussed above); we have addressed this comment in four ways:  1. We now include a statement in the ‘Key social processes’ section indicating that there was no significant difference in mean levels of trust between sites with and without social-ecological network closure, and we provide additional text explaining the relevance of finding higher variation in trust in sites without social-ecological network closure: “Although there were no significant differences in mean levels of trust between sites with and without social-ecological network closure, we found that there was significantly more variation in trust in both sites D and E compared to the other sites. This indicates that in sites D and E there is less agreement about whether others can be trusted, and the lack of social-ecological network closure in these sites suggests there may be pockets of mistrust – or at least a lack of trust – between resource competitors who do not communicate.” 2. We agree that it is possible that variation in fisher’s perceptions of the state of the resource could be due to more complex, or less obvious resource dynamics. We now explicitly discuss this alongside a more thorough discussion of our data limitations in the ‘Methods: Limitations’ section. 3. We now present fishing pressure alongside other potential factors that may be influencing biomass and functional richness across sites in Table 2, which shows that there is no significant difference in fishing pressure across sites with and without social-ecological network closure. Table S8 demonstrates that this result does not change even when you remove site A. 4. Perhaps most importantly, we now emphasize throughout our manuscript (e.g. in the introduction when we first introduce the mechanisms, and in our ‘Methods: Limitations’ section) that the mechanisms by which we suggest social-ecological network closure can support improved ecological conditions in this context have strong theoretical support. Thus, even though we lack direct empirical evidence of all the steps (and mechanisms) in the causal pathways linking social-ecological network closure to ecological conditions, we believe we are still well positioned to refer to the

In the Discussion, authors claim that “the benefit of the interdisciplinary social-ecological network approach described here is that it allows for a much more nuanced and precise understanding of the interdependencies between social and ecological components of ecosystems, allowing one to unpack the specific types of cooperative connections that facilitate or hinder effective action.”. Unfortunately, the examples above highlight that despite the great level of details in the social-ecological network analysis used in this study, serious methodological concerns remain and a tendency to emphasize particular cases to support hypothesized mechanisms undermine an otherwise interesting and original study. In summary, my view is that despite some exciting methodological and operational contributions (i.e., social- ecological network approaches applied to multi-resource commons), the main claim of this submission (that “closed social-ecological network structures amongst direct resource competitors facilitates more effective cooperation that can promote positive ecological outcomes in coral reefs”) is undermined by inferences that are cursory at best (due to analyses that are not especially rigorous or thorough), and hence unfortunately provides conclusions that are not particularly robust.	observed associations as driven by causation (further, we also wish to emphasise that how far arguments/claims of causation are taken is something that typically varies between scientific disciplines and research traditions, and within the social sciences, strong theoretical support usually gives more leverage to authors to refer to causality when interpreting and discussing the results from empirical analyses). However, we acknowledge that parts of our conclusions rest largely on assumptions, thus we have scrutinized our choice of words throughout the manuscript to better reflect this and are much more clear about the limitations of our data for identifying these mechanisms (‘Limitations’). Fore example, we caution that the results from our exploration of the potential mechanisms at play in this context in combination with the body of theory we rest our arguments on are only indicative that these mechanisms are in place (not conclusive), and that they should be seen as only one part of a triangulation effort meant to more thoroughly test our claims linking social-ecological network closure to ecological conditions.
Other comments The authors have done a very good job of limiting jargon, and presenting their results clearly. A little more detail in the text and/or in SI (in particular, general characteristics of each sites beyond those provided in SI; these may include location, governance arrangements, etc.) would help the reader evaluate fairly the research presented.	As described above, we have now added substantial detail concerning the sites, their location, and other aspects of the social-ecological context of this study in the form of additional figures, data, analyses, and text. We believe the paper is now a much stronger contribution, in part because of this review. For that, we thank you for the time and thought you put into this.
I found causal pathway described in this study overly simplistic. I am surprised that feedbacks from the ecological to the social systems (yet present in the SES framework this paper seems to rely on for the theoretical mechanisms) were not considered.	We agree that ecological-social feedbacks are incredibly important to consider. As such, we now include the following statement directly in the caption of Fig. 2: “It’s important to note that this figure is only illustrative of key mechanisms linking social-ecological network closure to ecological conditions, and does not include the full range of social-ecological interactions and feedbacks that can affect both ecological and social conditions in any given environmental system.” We also now explain how our framework can be expanded to explicitly capture them by drawing on dynamic/longitudinal data that tracks changes in social systems (and ecological systems) over time in the second to last paragraph of our discussion.
I don’t understand how and when the ecological network (representing trophic interactions among	Here our focus was largely on explaining the framework and demonstrating how it can be operationalized by

species) was used in this study. Please clarify its contribution to the study findings.	empirically testing a specific theoretical hypothesis. Our hypothesis focused on cooperation over shared species, so we held the ecological network constant in our network models. What that means is that its structure was taken as given, or fixed. We have clarified this for readers by amending the statement in lines 460-463, which now reads: “Because the focus here was on social processes, and particularly the propensity for fishers to form ties with direct resource competitors, the X and B level networks (Fig. 1) were fixed and treated as exogenous, which means that their structure was treated as given and therefore ties within these levels were not explicitly modeled.” To emphasize that our study represents a methodological advancement that will be used for further studies, we chose to include the ecological network data. We emphasize that future work could use this framework and even this specific ecological network data, which will be made available, to test explicit hypotheses about the formation of ecological links or other social-ecological interdependencies, which would explicitly model the ecological network given the social network, or model both the social and ecological network simultaneously (lines 341-347), and in the SI under the subheading ‘Multilevel Exponential Random Graph Models’).
Line 10: According to Table S1, 63 fishers were dropped from the analysis so the number here should be 693 instead of 756.	Thanks for drawing our attention to this. We have now adjusted the sample size to 648 to account for all individuals dropped from the analysis as well as the fishers who were associated with both sites A and D.

Reviewer 3	Author Response
This paper seeks to understand the relationship between multi-level network properties in socio-ecological systems and its relationship to ecosystem viability using ERGMs. This is an interesting approach that directly links natural elements of ecosystems with human behavior. The approach is novel but there needs to be more background information to help in interpreting the statistical findings of the study. It would be nice to know the spatial distribution of the reef systems and the spatial and community composition of the fishers (e.g., maps might help). To get a better understanding of the relationship between landing sites and reefs a 2-mode matrix involving reefs and landing sites was created from data in the supplemental materials. Basically, based on the ecological assessments sites A, B and C appear to be more ecologically viable while reefs D and E are less so. Correspondingly, there is multi-level network closure in the networks of the viable systems but not in the other two reefs. The attached graph is the two mode network of reefs by landing sites. What is interesting is	In response to this comment and those of Reviewers 1 & 2, we have now added substantial detail concerning the sites, their location, and other aspects of the social-ecological context of this study in the form of additional figures, data, analyses, and text. We believe the paper is now a much stronger and that the statistical findings of the study are more clear. We further addressed this comment in three ways:  1. We now include a map showing the spatial distribution of sites (Fig S1). 2. Thanks for including the 2 mode matrix. It helped us to realize that we needed to explain more clearly the distinction between our sites. In the beginning of the SI, we now explain that sites D and A are in close proximity to each other (4.5 km apart), and thus use the same landing sites. However, these communities and their fishing locations are distinct (located 4.5 km apart). However, while conducting fieldwork we found that a

that reef E, one of the less viable reefs, clearly uses very different landing sites. However, reefs A and D are essentially structurally equivalent in the graph yet one reef system is more viable than the other. It appears that the fishers at the two reefs use the same landing sites, yet have separate networks. Further background information may help in better clarifying this observation. The question however, is given the similar behaviors in choices of landing sites between the fishers at the two reefs, what kinds of interactions occur between the fishers from the different reefs? Without this explanation it is difficult to interpret the study's findings.

Also, fishers around the world tend to engage in targeting multiple species with multiple gears. Do fishers in this context only use one type of gear when targeting species or do they engage in switching behaviors throughout the year? This needs clarification.

minority of respondents fished in both of these locations (n=45), and considered themselves part of both of these fishing communities. As discussed above in response to Reviewer 2, we accounted for this directly in our original analysis by including them in both community networks. We made this decision because they identified themselves as part of both communities, and they fish in both locations. They therefore can have an effect on our measures of ecological conditions in both sites, and whether or not they form cooperative communication ties with others and target different species within each site is important to consider. Our qualitative insights from the on-the-ground data gathering efforts indicated that these fishers adjust their social behaviours depending on the context (i.e., which site they are operating/fishing in), which is consistent with research in social psychology (e.g., Turner et al. 1994 *Personality & Social Psych Bulletin*). However, during our revisions, we ran our network models for sites A and D again with these fishers removed as a sensitivity analysis, and found no meaningful difference in our results (i.e., all parameters had the same sign and level of significance in the models, and the models showed no difference in goodness-of-fit). This shows that these multi-site fishers alone are not significantly different from others in these communities. Importantly, they are not driving the tendency for or against social-ecological network closure; or for it in one site, and against it in another.

3. Thanks for the chance to add more clarification about fishing gears. As you correctly pointed out, many fishers around the world tend to switch gears throughout the year depending on the season. However, in these sites, most fishers only used one primary gear type throughout the year. We now include more detail about this in the SI under the subheading "Social-Ecological Ties", which states: "Though switching gears throughout the year depending on the season is quite common in many fisheries around the world, the majority of fishers in our sample used only one primary gear type year-round. Social-ecological ties were therefore identified by linking individual fish species to individual fishers via their primary fishing gear as identified by fishers in our fisher survey (Table S4)."

Reviewers' Comments:

Reviewer #1:

Remarks to the Author:

The authors have done a nice job addressing my comments, and I have no further concerns.

Reviewer #2:

Remarks to the Author:

I think the social-ecological network closure approach used in this study is good and has a lot of potential application in quantifying cooperative communication relationships in the commons. However, I think the authors are let down by the design and (some of the) data in this study, which does not allow to support their main claim ('when fishers facing dilemmas form cooperative communication ties with direct resource competitors, they can achieve positive gains in both reef fish biomass and functional richness'). While the authors have responded to my previous concerns to some extent, I feel many of the same arguments apply.

The authors argue that they have got around the problems of confounding factors. Unfortunately, I don't think this is the case. For example, it is not a surprise that the t-tests are not significant when comparing biophysical conditions between closed vs. open network given the extremely small sample size (and high variability). A power analysis would probably show that it is almost impossible to detect significant differences (strong probability of type II error). On a similar note, the distribution of the biomass/functional richness data, it is very plausible that the assumption made when performing a t-test were not met. Those assumptions are clearly not met in the other tests (e.g., when Levene test is significant, it means a t-test is not suitable, yet it was used in the case of 'trust'), which casts doubts on the statistical rigor of this critical part of the analysis. Another factor that could explain the absence of significant differences in biophysical characteristics across groups is the quality of the data used in input: coral cover and fish data were not collected at the same time (6 years apart in one case). I couldn't find any information of the habitat beyond depth (but at equal depth and coral cover, reef flats and slopes are very different I suppose). Socioeconomic (e.g., use of a freezer, feet motorization, etc.) and institutional factors (e.g., clearly defined boundaries, history of use, etc.) that are known to drive fish assemblages are also missing. Together, these issues undermine this study's ability to test the central hypothesis (social-ecological network closure is associated with positive resource condition) or to provide any empirical preliminary insight into the mechanisms involved, even if these mechanisms have been theorized.

The sensitivity analyses (2) and (3) in Table S8 do not provide information about among-group differences in biophysical variables, although I suspect the results would not be affected for the reasons described above.

Although the authors have been more explicit about the assumptions and limitations of their study, sentences in the abstract (lines 45-48), results (lines 226-229 and 235), discussion (lines 270-272 and 356-259), Limitations section (line 521) and even the title clearly lack the nuance required. These bold statements are misleading (see reasons above), and this is particularly concerning given that some management recommendations are proposed at the end of the Discussion. In making recommendations about interventions, it is important that there is strong support for the hypothesis, and this is not the case here. Authors can refer to Game et al.'s comment in Nature Sustainability for a definition of what I mean by (lack of) 'strong evidence'.

I am not convinced by the authors' empirical argument about trust being mediated by network closure

(i.e., cooperation). Again, means do not differ across groups as it was expected, and authors focus instead on the unequal variance, suggesting that this could be due to 'pockets of mistrust'. I would argue that this might as well be due to 'pockets of high trust'. Clearly the empirical evidence supporting the expected role of network closure in building trust is weak and should not be highlighted as this is the case here.

The contrast between the level of sophistication of the network approach and the overly simplistic approach to test the hypothesis is surprising, and I would advise authors to reframe their paper around the strong aspect of their study (quantifying cooperative cooperation ties using multilevel exponential random graph modelling).

Other minor comments:

- Typos in lines 139 ('to affect'), 291 ('development of trust?') and 814 ('are reported?')
- I don't understand why degrees of freedom differ in t-test on fishable biomass vs. functional richness.
- Despite authors' response, I still don't understand when the ecological network based on trophic interactions was used. My understanding is that none of the indices presented in this study rely on this ecological network.

Reference:

Game, E.T., Tallis, H., Olander, L., Alexander, S.M., Busch, J., Cartwright, N., Kalies, E.L., Masuda, Y.J., Mupepele, A.-C., Qiu, J., Rooney, A., Sills, E., Sutherland, W.J., 2018. Cross-discipline evidence principles for sustainability policy. *Nat. Sustain.* 1, 452–454. doi:10.1038/s41893-018-0141-x

Reviewer #3:

Remarks to the Author:

The authors have addressed my major concerns. One comment related to the following from the supplemental materials. They state: "However, because fishing gears helped to define the social-ecological ties in the present study, gear-based effects may to some extent help to explain why fishers who target the same resource chose to form cooperative communication ties in some of our sites, which has important practical implications. Indeed, the value of exchanging experiences and knowledge with others using the same technology is rather intuitive – it allows actors to accrue technical knowledge that can help them to maximize harvest levels and operate more efficiently. Yet because actors using the same technology also tend to target the same resource for extraction, these exchanges can also enhance trust and a shared ecological understanding of factors important for the resource to be sustained." Many conflicts in fisheries around the world are gear conflicts often involving the same species. This is particularly the case for fixed gear versus mobile gear targeting the same species.

Finally, it would have been nice to have more comprehensive measures of trust and the perceptions of ecosystem health. With a series of questions on individual perceptions of ecosystem health a more robust assessment of consensus could have been achieved and would have allowed for a better statistical test of the hypothesis.

Reviewer #2 Main Comments	Author Response
I think the social-ecological network closure approach used in this study is good and has a lot of potential application in quantifying cooperative communication relationships in the commons. However, I think the authors are let down by the design and (some of the) data in this study, which does not allow to support their main claim ('when fishers facing dilemmas form cooperative communication ties with direct resource competitors, they can achieve positive gains in both reef fish biomass and functional richness'). While the authors have responded to my previous concerns to some extent, I feel many of the same arguments apply. The authors argue that they have got around the problems of confounding factors. Unfortunately, I don't think this is the case. For example, it is not a surprise that the t-tests are not significant when comparing biophysical conditions between closed vs. open network given the extremely small sample size (and high variability). A power analysis would probably show that it is almost impossible to detect significant differences (strong probability of type II error).	We realize now that there may have been some confusion over the number of samples used to evaluate biophysical conditions (i.e., coral cover and rugosity), which we claim responsibility for. While there were 5 study locations, our sample of biophysical conditions (i.e., coral cover and rugosity) was comprised of a total of 71 individual transects. This is clearly noted in Table S7, where we report all our data sources. However, this was not clear in tables Table 2 and S8 of our previously revised manuscript, where we reported the results of our comparison in means. We have amended this in our revised manuscript to prevent any further confusion. In addition to the t-test for equality of means we conducted using this sample of 71 transects, which indicates that there is no significant difference in these conditions across sites with and without social-ecological network closure, we also included effect size estimates and confidence intervals (Table 2). These clearly show that there is no meaningful difference in these factors, i.e.,

	the effect size estimates are small (-0.23 for coral cover) or virtually non-existent (-0.01 for rugosity), and their confidence intervals both overlap zero (-0.64, 0.18 and -0.41, 0.40 respectively). A post-hoc power analysis would not provide any additional insight beyond the results of these tests, as observed power is a function of the p-value of the test for equality of means. With such non-significant effects, one will always have low power (Goodman and Berlin 1994 Annals of Internal Medicine, Hoenig & Heisey 2001 The American Statistician, Lenth 2001 The American Statistician). This does not negate our ability to make the inferences we have on the data, which here suggested there is little to no effect. Furthermore, we report the same metrics for our indicators of ecological condition (fish biomass and functional richness, Fig. 3), and we do find differences in sites with and without social-ecological network closure that are both significant (p-value) and meaningful (effect size). We are therefore confident in our claim that that we have found evidence that ecological conditions are different across sites, but that biophysical conditions are not. The difference in ecological conditions is thus unlikely to be due to variation in biophysical conditions, as we state in lines 185-187, and there is no new information in a post-hoc analysis.
On a similar note, the distribution of the biomass/functional richness data, it is very plausible that the assumption made when performing a t-test were not met.	The assumptions of our tests were met. For some of our variables, such as biomass, the variance was not equal across groups. We therefore used Satterthwaite's approximation formula (1946) to approximate the degrees of freedom, which accounts for unequal variance. The results of these tests are identical whether you use this approximation or perform the Mann-Whitney U non-parametric test for equality of means, which does not require the traditional assumptions of parametric tests, such as normality and equal variances. Our results are thus robust to different approaches for testing for equality of means. Initially, we did not go into this

Those assumptions are clearly not met in the other tests (e.g., when Levene test is significant, it means a t-test is not suitable, yet it was used in the case of 'trust'), which casts doubts on the statistical rigor of this critical part of the analysis.	level of detail in our manuscript, but we have now made this explicitly clear in lines 449-450, lines 460-472, in the footnote on page 9, and in the footnotes of Table 2 and S8. We did not use a t-test for trust. We used a Mann-Whitney non-parametric test, which does not assume equal variance. The results of this test, which indicate there is no difference in mean levels of trust between sites with and without social-ecological network closure, are identical whether you use this test, the t-test with Satterthwaite's approximation for degrees of freedom (which also accounts for unequal variance), or a linear mixed model with a fixed effect for group and a random effect for individual [which accounts for the minority of individual fishers (n=45 out of 648) whose measures of trust were not independent because they identified themselves as part of two of our study communities]. We did not go into this level of detail in our manuscript because our examination of trust was not our primary focus, and because these are relatively simple statistical procedures. However, we see now that providing this detailed information would certainly help to prevent any doubt about the statistical rigor of this (and any other) result we have reported. We therefore now include this detail in lines 459-460 and in the footnote that appears on the same page. We also provide similar detail regarding our test for equality of means in regards to participation in decision making in the footnote appearing on pg. 22.
Another factor that could explain the absence of significant differences in biophysical characteristics across groups is the quality of the data used in input: coral cover and fish data were not collected at the same time (6 years apart in one case).	There appears to be a misinterpretation of our description of the data analyses, specifically the use of data spanning multiple years. Critically, the primary analyses presented in the main body of our manuscript includes coral cover and fish data that were

I couldn't find any information of the habitat beyond depth (but at equal depth and coral cover, reef flats and slopes are very different I suppose).

Socioeconomic (e.g., use of a freezer, feet motorization, etc.) and institutional factors (e.g., clearly defined boundaries, history of use, etc.) that are known to drive fish assemblages are also missing.

collected at a minimum of two years apart in each site. Importantly, in the absence of a major disturbance, which we have ruled out, coral cover has been shown to be relatively stable across short time frames such as these (including in Kenya – e.g. Darling et al. 2010 Conservation Letters). Our only analyses that compare data collected up to 5 years apart (not 6 as suggested) is in a supplemental analysis we included as a sensitivity test (see Table S7, S8). Importantly, this supplementary analysis does not negate the findings of our primary analysis presented in the main paper.

We provided two quantitative measures of habitat in our revised manuscript: coral cover and rugosity (Table 2). Rugosity is known to be one of the biggest habitat drivers of reef fish composition and biomass (reviewed by Graham & Nash 2013, *Coral Reefs*), so the fact that we have found no difference in rugosity rules out one of the main potential confounding factors. In addition, in the methods section we specifically described the habitat of each site: “All fishing areas sampled were shallow (<10m depth), exposed to similar environmental conditions (Table 2) and have a similar disturbance history (e.g., coral bleaching).”

In their initial review, this reviewer highlighted the importance of the following specific potentially confounding factors: **depth, habitat, age of the organization, history of disturbance, proximity to other human disturbances**. We included all of these factors in our revised manuscript with the exception of age of the organization (which we did not include because we were not studying organizations). Yet now a new list of factors has been identified.

Still, we are more than happy to include this final list of factors in a revised manuscript if necessary, which will show that there is no

	difference across sites in use of fish freezers, fleet motorization, clearly defined boundaries, and/or history of use.
The sensitivity analyses (2) and (3) in Table S8 do not provide information about among-group differences in biophysical variables, although I suspect the results would not be affected for the reasons described above.	We did not provide information about among-group differences because the focus of our study was specifically focused on estimating the potential effect of social-ecological network closure.
Although the authors have been more explicit about the assumptions and limitations of their study, sentences in the abstract (lines 45-48), results (lines 226-229 and 235), discussion (lines 270-272 and 356-259), Limitations section (line 521) and even the title clearly lack the nuance required. These bold statements are misleading (see reasons above), and this is particularly concerning given that some management recommendations are proposed at the end of the Discussion. In making recommendations about interventions, it is important that there is strong support for the hypothesis, and this is not the case here. Authors can refer to Game et al.'s comment in Nature Sustainability for a definition of what I mean by (lack of) 'strong evidence'.	We strongly disagree with this assessment given that the doubt concerning our analyses and the statistical rigor of our approach is driven by a few key misunderstandings, as described above. Still, we have tempered the language even further in our revised manuscript in several of the places highlighted. For example, we have changed the title of our manuscript to “Social-ecological alignment and ecological conditions in coral reefs”, and changed the language discussing our results in the abstract and discussion, e.g., lines 239-241 (270-272 in the reviewer comments) now read (underline for emphasis): “Our quantitative and qualitative results provide evidence that closed social-ecological network structures amongst direct resource competitors may facilitate more effective cooperation that is associated with positive ecological conditions in coral reefs.” We have also carefully read through the remainder of our manuscript to ensure that the language is accurate and sufficiently cautious where necessary. Regarding our management recommendations, these focus on building community capacity which is not novel. However, our focus on building gear-based dialogues is one that has direct applicability to the problems of overfishing. Our recommendations are based on the evidence presented in the manuscript that show unequivocally that communities of practice exist around specific gear and technology

	usage, and that there is variability in ecological performance that is associated with the structure of relationships among these fishers and their resources. Further - our management recommendations to support the development of social ties and relationships among resource users are supported by a prodigious literature on common-pool resources as well as a wealth of practitioner knowledge (two of our authors are conservation practitioners, and know this space extremely well).
I am not convinced by the authors' empirical argument about trust being mediated by network closure (i.e., cooperation). Again, means do not differ across groups as it was expected, and authors focus instead on the unequal variance, suggesting that this could be due to 'pockets of mistrust'. I would argue that this might as well be due to 'pockets of high trust'. Clearly the empirical evidence supporting the expected role of network closure in building trust is weak and should not be highlighted as this is the case here.	We would like to reiterate that our empirical examination of trust was not the primary focus of this paper. The primary focus of this paper was to highlight the potential of the social-ecological network framework for capturing complex relationships between people and nature, and to apply this framework by testing the role of social-ecological network closure on ecological conditions. As we have made clear throughout our revised manuscript, our empirical analysis of trust was merely a preliminary empirical exploration of one of the potential mechanisms by which social-ecological network closure may impact ecological conditions. Thus, our main finding (social-ecological network closure is associated with better ecological conditions) is not contingent upon trust. Rather, the expected mediating effect of trust represents only one of several potential mechanisms underpinning this observed relationship (Fig. 2). In regards to our preliminary empirical examination of trust, in our manuscript we clearly explained that the role of network closure in building trust has an incredibly strong history of theoretical and empirical evidence across many different contexts (lines 66-69). Thus, as suggested, we did expect to find higher levels of trust in communities with social-ecological network closure than without – and we were very explicit in our manuscript that we did not find empirical evidence of this, i.e., in our

previously revised manuscript, lines 205-206 read: “..there were no significant differences in mean levels of trust between sites with and without social-ecological network closure.”

Still, the fact that sites without social-ecological network closure have significantly higher variation in trust is equally important, and should not be swept under the rug. As we stated in our previously revised manuscript, this indicates there is less agreement about whether people can be trusted (line 208) - which indeed may be indicative of both pockets of trust *and* mistrust.

To clarify and contextualize these results even further, we have now amended this discussion to read (lines 205-206):

“Importantly, we did not find evidence of that mean levels of trust differed between sites with and without social-ecological network closure. However, we did find that there was significantly more variation in trust in both sites D and E compared to other sites. This indicates that in sites D and E there is less agreement about whether others can be trusted, and the lack of social-ecological network closure in these sites suggests there may be pockets of mistrust – or at least a lack of trust – between resource competitors who do not communicate²⁸. On the flip side, there may also be pockets of trust.”

In our ‘Limitations’ section, we specifically caution that the results from our exploration of the potential mechanisms at play in this context (e.g. trust) in combination with the body of theory we rest our arguments on are only indicative that these mechanisms are in place, not conclusive, and given their strong theoretical support, they should be seen as only one part of a triangulation effort meant to more thoroughly test our claims linking social-ecological network closure to ecological conditions.

The contrast between the level of sophistication of the network approach and the overly simplistic approach to test the hypothesis is surprising, and I would advise authors to reframe their paper around the strong aspect of their study (quantifying cooperative cooperation ties using multilevel exponential random graph modelling).	We strongly disagree with this statement. We have provided a substantial amount of information to support our hypothesis that social-ecological network closure is associated with better ecological conditions, including data that clearly indicates there is indeed a significant and meaningful difference in ecological conditions in sites with and without social-ecological network closure (Table 1, Fig. 3). In addition, we provide substantial and diverse information on potentially confounding factors, including data on biophysical conditions, environmental conditions, and human impact characteristics (Table 2), and on social and institutional conditions (Table 4). Finally, we provide preliminary insight into the mechanisms by which social-ecological network closure may impact ecological outcomes (Fig. 2, Table 3), which is backed by a long history of theoretical and empirical research.
Reviewer #2 Minor comments	Author Response
Typos in lines 139 ('to affect'), 291 ('development of trust?') and 814 ('are reported?')	We have fixed these typos.
I don't understand why degrees of freedom differ in t-test on fishable biomass vs. functional richness	As discussed above, Satterthwaite's approximation was used to estimate the degrees of freedom to account for unequal variance.
Despite authors' response, I still don't understand when the ecological network based on trophic interactions was used. My understanding is that none of the indices presented in this study rely on this ecological network.	As we state in lines 460-463, because the focus here was on social processes, and particularly the propensity for fishers to form ties with direct resource competitors, the ecological network was fixed and treated as exogenous in our analysis, which means that its structure was treated as given and therefore the ecological network was not explicitly modelled in our empirical analysis. To emphasize that our study represents a methodological advancement that will be used for further studies, we chose to include the ecological network data even though we did not explicitly model these ties in our analysis. We emphasize that future work could use this framework and even this specific ecological network data, which will be made available, to test explicit hypotheses

	about the formation of ecological links or other social-ecological interdependencies, which would explicitly model the ecological network given the social network, or model both the social and ecological network simultaneously (lines 341-347).
--	--

Reviewer #3 Comments	Author Response
The authors have addressed my major concerns. One comment related to the following from the supplemental materials. They state: "However, because fishing gears helped to define the social-ecological ties in the present study, gear-based effects may to some extent help to explain why fishers who target the same resource chose to form cooperative communication ties in some of our sites, which has important practical implications. Indeed, the value of exchanging experiences and knowledge with others using the same technology is rather intuitive – it allows actors to accrue technical knowledge that can help them to maximize harvest levels and operate more efficiently. Yet because actors using the same technology also tend to target the same resource for extraction, these exchanges can also enhance trust and a shared ecological understanding of factors important for the resource to be sustained." Many conflicts in fisheries around the world are gear conflicts often involving the same species. This is particularly the case for fixed gear versus mobile gear targeting the same species.	This is an important comment. Indeed, we discuss this issue in some detail in lines 284-300 of the main body of our manuscript, which reads: “Yet given the competitive nature of many common-pool resource systems such as reef fisheries⁵⁵, important questions remain regarding how these relationships can be built. Here, key social-ecological interactions were defined as those that linked fishers to specific species based on their fishing gear (Fig. 1). Our results thus suggest that stimulating gear-based communication may indirectly lead to a greater propensity for social-ecological network closure since the same set of species tend to be targeted by the same gear (Table S3, SI). These communication channels can be facilitated by creating communities of practice centered around gear and technology, which can act to stimulate learning, build trust, and enhance shared ecological understanding of factors important for resources to be sustained⁵⁶. However, caution is warranted, as efforts to build such communities of practice could lead to the emergence of competing gear-based coalitions and a zero-sum game where the potential ecological benefits from restricting one gear are captured by users of another gear³⁶. This is a genuine risk in multi-species, multi-gear reef fisheries and other similar common pool-resource systems, where gear competition is ubiquitous. Thus, broader community building strategies that seek to establish communication and trust across all direct resource competitors, including actors

Finally, it would have been nice to have more comprehensive measures of trust and the perceptions of ecosystem health. With a series of questions on individual perceptions of ecosystem health a more robust assessment of consensus could have been achieved and would have allowed for a better statistical test of the hypothesis.

using different gear types but overlapping in target species, is critical for achieving long-term sustainability.”

In principle we agree that it would have been nice to have more data on these complicated social processes; however, it simply wasn't practically feasible to collect multiple indicators of all of the social, institutional, and economic factors and conditions included in this research because of high risk of respondent fatigue. We specifically highlight this in lines 499-503.

Thankfully, our hypothesis and main finding (social-ecological network closure is associated with better ecological conditions) is not contingent upon these measures. We were very careful to explain in our revised manuscript that our exploration of trust and perceptions of ecosystem health were merely a preliminary empirical exploration of some of the potential mechanisms by which social-ecological network closure may impact ecological conditions. For example, after introducing our hypothesis and how we go about testing it, we now state (lines 103-105):

“We also conducted a preliminary assessment of indicators of the key social processes supported by social-ecological network closure (Fig. 2) across sites to explore whether they aligned with our theoretical expectations.”

Reviewers' Comments:

Reviewer #1:

Remarks to the Author:

No additional comments from earlier review

Reviewer #3:

Remarks to the Author:

An earlier comment concerned other measures of trust. I understand the problems with respondent fatigue but it would be useful to see the relationship between their simple measure of trust, based on a Likert scale, and network closure, often used as a network measure of trust. With the exception of this the authors have basically addressed my concerns.

Reviewer #4:

Remarks to the Author:

The massive sustainability challenges we face today call for new research approaches that can analyse the integrated dynamics of human and natural dimensions. This paper builds on, and makes substantial progress, on recent interdisciplinary research on complex networks. Specifically it sets out to empirically test work on social-ecological network motifs, applies new exponential graph modelling on an extremely detailed data-set (despite the low n) and unpacks the ways in which societies (small-scale fishing communities in Kenya) and nature (adjacent coral reef fish communities) are interdependent, and empirically links this to specific social mechanisms that build on solid theoretical ground.

I have seen that the authors have taken huge steps in improving the paper based on previous rounds of reviewer comments. This shines through. It is an elegantly written paper with nice visuals, and accessibly presented results, which makes it easy to follow despite the complex subject. The conclusions are well-supported and the novelty is very high.

The theoretical basis presented in Figure 2. are well-founded and quite uncontroversial. There are plenty of studies that highlight how the features represented by social-ecological closure (communication and cooperation) lead to increased levels of trust, common visions, better conflict resolution and agreements on the rules a community abides by. Similarly there are strong evidences of such elements facilitate collective action and lead to positive resource outcomes.

Most importantly, the paper does a painstaking job of accounting for other relevant biophysical, environmental, human impact (Table 2), social (Table 3) and institutional (Table 4) drivers of environmental conditions. The past decade have seen massive progress being made in unearthing the key biophysical (e.g. rugosity, SST), human proximate (fishing pressure), human distal (market gravity), and institutional (e.g. Ostroms design principles) drivers that affect the condition of coral reefs. The paper shows that differences in these above mentioned drivers cannot explain the differences in fish biomass and functional diversity. Crucially, the importance of social-ecological network closure continues to shine through as the key determinant of fish biomass and functional diversity.

I don't have any minor comments, since most niggles and previous weaknesses seem to have been ironed it in previous iterations of this paper.

REVIEWERS' COMMENTS (**Author response in bold**):

Reviewer #1 (Remarks to the Author):

No additional comments from earlier review

NA

Reviewer #3 (Remarks to the Author):

An earlier comment concerned other measures of trust. I understand the problems with respondent fatigue but it would be useful to see the relationship between their simple measure of trust, based on a Likert scale, and network closure, often used as a network measure of trust. With the exception of this the authors have basically addressed my concerns.

Our measure of trust is highly related to network closure. Estimates of both social and social-ecological network closure across all sites can be derived from our exponential random graph model results in Fig. 3. These results demonstrate that the former (social network closure) is positively significant across all sites, while the later (social-ecological network closure) is positively significant in sites A-C (but not D-E). (Please note that the magnitude of within level parameter estimates from multilevel ERGMs cannot be directly compared to across-level parameter estimates; in other words, one cannot directly compare the parameter for social network closure to the parameter for social-ecological network closure). Results of our measure of trust appear in Table 2. These demonstrate that we do not see any differences in mean level of trust across sites (which directly corresponds with our results regarding social network closure), yet we do see a difference in the variation (i.e., higher levels of variation in sites D-E, which directly corresponds to our results regarding social-ecological network closure).

Reviewer #4 (Remarks to the Author):

The massive sustainability challenges we face today call for new research approaches that can analyse the integrated dynamics of human and natural dimensions. This paper builds on, and makes substantial progress, on recent interdisciplinary research on complex networks. Specifically it sets out to empirically test work on social-ecological network motifs, applies new exponential graph modelling on an extremely detailed data-set (despite the low n) and unpacks the ways in which societies (small-scale fishing communities in Kenya) and nature (adjacent coral reef fish communities) are interdependent, and empirically links this to specific social mechanisms that build on solid theoretical ground.

I have seen that the authors have taken huge steps in improving the paper based on previous rounds of reviewer comments. This shines through. It is an elegantly written paper with nice visuals, and accessibly presented results, which makes it easy to follow despite the complex subject. The conclusions are well-supported and the novelty is very high.

The theoretical basis presented in Figure 2. are well-founded and quite uncontroversial. There are plenty of studies that highlight how the features represented by social-ecological closure (communication and cooperation) lead to increased levels of trust, common visions, better conflict resolution and agreements on the rules a community abides by. Similarly there are strong evidences of such elements facilitate collective action and lead to positive resource outcomes.

Most importantly, the paper does a painstaking job of accounting for other relevant biophysical, environmental, human impact (Table 2), social (Table 3) and institutional (Table 4) drivers of environmental conditions. The past decade have seen massive progress being made in unearthing

the key biophysical (e.g. rugosity, SST), human proximate (fishing pressure), human distal (market gravity), and institutional (e.g. Ostroms design principles) drivers that affect the condition of coral reefs. The paper shows that differences in these above mentioned drivers cannot explain the differences in fish biomass and functional diversity. Crucially, the importance of social-ecological network closure continues to shine through as the key determinant of fish biomass and functional diversity.

I don't have any minor comments, since most niggles and previous weaknesses seem to have been ironed in previous iterations of this paper.

Thank you for your highly positive comments.